# Evidence of direct and indirect reciprocity in network-structured economic games
Daniel Redhead [1,2,3,5] ✉, Matthew Gervais [4], Kotrina Kajokaite[1], Jeremy Koster[1], Arlenys Hurtado Manyoma[1], Danier Hurtado Manyoma[1], Richard McElreath [1] & Cody T. Ross [1,5] ✉

Formal theoretical models propose that cooperative networks can be maintained when individuals condition behavior on social standing. Here, we empirically examine the predictions of such models of positive and negative indirect reciprocity using a suite of network-structured economic games in four rural Colombian communities ($N_{ind}$ = 496 individuals, $N_{obs}$ = 53,876 ratings/transfers). We observe that, at a dyadic-level, individuals have a strong tendency to exploit and punish others in bad standing (e.g., those perceived as selfish), and allocate resources to those in good standing (e.g., those perceived as generous). These dyadic findings scale to a more generalized, community level, where reputations for being generous are associated with receipt of allocations, and reputations for being selfish are associated with receipt of punishment. These empirical results illustrate the roles that both positive and negative reciprocity, and costly punishment, play in sustaining community-wide cooperation networks.

Humans have an exceptional ability to cooperate at large scales[1]. This human propensity to cooperate widely, often with genetically-unrelated peers, was once seen as a fundamental puzzle in the evolutionary social sciences[2]. Kin selection models[3,4] were quick to demonstrate that cooperation—defined as incurring a fitness cost to provide a fitness benefit to others[5]—can be maintained by inclusive fitness considerations broadly[6], and kin detection and discrimination more specifically[7]. However, such mechanisms were not able to explain the wide range of cases in which humans cooperate with non-kin, as measured both by economic games [e.g.,[8–10]] and social network approaches [e.g.,[11–24]]. Although most network-based studies of cooperation find that kinship is an important predictor of cooperation, there is ample evidence that other mechanisms—such as reciprocity[25] and reputation or status differentiation[26,27]—are similarly important factors in determining the structure of cooperative networks. In short, cooperative networks are substantially broader than kinship networks.

As with early models of kin selection, models of direct reciprocity[2,28,29] appeared well positioned to explain how cooperative networks are produced and maintained. Reciprocity with non-kin has been documented in numerous human studies [e.g.,[11,12,14–21,24]], and in non-human animals [e.g.,[30]]—though debates over what constitutes evidence of reciprocity continue[31].

In the context of human social networks, reciprocal exchange need not be based on strict, Tit-for-Tat behavioral responses; instead, inter-personal sentiments may track the fitness affordances of certain relationships, and translate perceptions about fitness-relevant characteristics of potential partners (e.g., their selfishness versus generosity) into behaviors (e.g., cooperation, exploitation, or punishment;[32,33]; see also[34,35]). When two individuals (i.e., a dyad) view each other as valuable social partners (e.g., non-selfish and competent), they are expected to maintain cooperative relationships even if transient resource insecurities preclude direct reciprocation, or if implementation errors lead to unintended defections. Formal models of such dynamics re-envisage reciprocity using standing-conditional rules, rather than absolute rules [e.g., through strategies like "contrite Tit-for-Tat" or "arbitration Tit-for-Tat";[36–38]].

Regardless of the details, models of direct reciprocity appear insufficient to explain the breadth of human cooperation, and have been found to support the evolution of cooperation mostly in narrow, abstract experimental conditions [e.g., where individuals are constrained to only one fixed partner in iterative games;[39]]. Moreover, the scope of reciprocal cooperation becomes restricted as group size increases[40]. Cooperation in humans is routinely observed between strangers in fleeting and transient interactions, without any need for (or expectation of) direct reciprocation[41]. So, although inclusive fitness and direct reciprocity are important factors in sustaining

[1]Department of Human Behavior, Ecology and Culture, Max Planck Institute for Evolutionary Anthropology, Leipzig, Germany. [2]Department of Sociology, University of Groningen, Groningen, The Netherlands. [3]Inter-University Center for Social Science Theory and Methodology (ICS), University of Groningen, Groningen, The Netherlands. [4]Division of Psychology, Department of Life Science, Brunel University, London, UK. [5]These authors contributed equally: Daniel Redhead, Cody T. Ross. ✉e-mail: daniel_redhead@eva.mpg.de; cody_ross@eva.mpg.de

cooperation, it seems that other mechanisms are still needed to explain the breadth of cooperation observed in many human groups.

For instance, third-party (or institutionalized) punishment of non-cooperators can stabilize large-scale cooperation[42,43]. In such models, individuals who observe a group member being wronged may pay a cost to inflict a higher cost upon a wrong-doer [i.e., the 'free-rider' or 'defector';[44]]. Finding themselves at the receiving end of such punishment, wrong-doers may feel incentivized to alter their behavior (e.g., to cooperate more) in order to avoid future punishment. In the case of institutionalized punishment, members of a group allocate a share of resources to reward a select set of individuals for punishing non-cooperators [e.g., state-instituted police are provided monetary salaries;[45]]. However, a dilemma emerges in the absence of direct rewards for punitive acts: the optimal strategy for third-party individuals is to let others pay the costs of punishing non-cooperators [i.e., there is a "second-order free-rider problem";[46]].

A solution to the second-order free-rider problem—and an independent mechanism for the evolution of large-scale cooperation more generally[47–49]—is indirect reciprocity[50]. In models of indirect reciprocity, individuals engage in costly cooperation or punishment in order to uphold their own standing (i.e., their dyadic standing based on first-person knowledge of past behavior) or reputation (i.e., their social standing based on aggregate third-party accounts and gossip). In such models, an individual's standing/reputation is assumed to depend on past behavior—which itself is generally assumed to be publicly observable and accurately recalled by group members[51]. Cooperative acts improve an individual's reputation, while non-cooperative acts have deleterious impacts on reputation. This linkage between behavior and reputation generates incentives to both cooperate and punish, as the reputation of an individual can be affected both by their baseline behavior and by how they respond to defectors. Given such an incentive structure, indirect reciprocity has been shown to effectively stabilize both cooperation and punishment across some theoretical conditions[50,52,53] and empirical settings[54–56].

There are, however, multiple forms of indirect reciprocity. Initially, theoreticians focused on positive indirect reciprocity—a process in which individuals condition their *cooperative behavior* on the reputation or social standing of others, with those in good standing being more likely to receive help when in need[49]. Negative indirect reciprocity captures a similar process where individuals base their decisions to *exploit* others on the reputation or social standing of those others[57,58].

Formal models indicate that positive indirect reciprocity stabilizes cooperation across a variety of conditions, but the initial set of assumptions required to derive these models demand that populations are to some degree harmonious a priori[49,58] and have effective networks of communication[59]. Recent models have attempted to deal with these issues, arguing that targeted exploitation has helped to launch human cooperation, without requiring social harmony a priori[58]. Specifically, Bhui et al.[58] show that preferential exploitation of those with selfish reputations can create an incentive structure that facilitates the emergence and maintenance of cooperation. If exploitation is both reputation-conditional and sufficiently costly—and if cooperative behavior can improve reputation—then individuals may be incentivized to cooperate in order to improve their reputations and minimize their chances of becoming targets of exploitation.

Currently, few empirical studies have attempted to test how networks of cooperation, exploitation, and punishment in non-WEIRD contexts[60] are simultaneously structured by dyadic perceptions of inter-personal social standing and community-wide reputations [but see refs. 61–64, for relevant case studies]. In the current study, we present an integrated approach for studying the multiplex (i.e., overlapping or interacting) structure of economic game-play and perceptual networks. In doing this, we empirically examine the roles that positive and negative reputations play in guiding people's decisions to cooperate with, exploit, and punish others. To do this, we conducted a set of three community-wide network-structured economic games [i.e., the RICH games[63]] in four rural Colombian communities. These games measure individuals' preferences for cooperating with, exploiting, and punishing individuals in their communities[20]. The games are *recipient*

*identity-conditioned*—i.e., individuals know the identity of others during game play—but remain *decider-confidential*. In other words, individual $j$ does not know—when rating individual $k$ as selfish or generous—who individual $k$ gave to, exploited, or punished. We pair the game data with dyadic peer ratings of selfishness and generosity to study how perceptions of social standing are associated with game-play behavior. Therefore, our measures of social standing/reputation are not based on game-play, but rather reflect individuals' pre-existing perceptions of other group members.

If direct positive and negative reciprocity play a role in structuring economic game-play networks, then we would expect that individuals act upon their own personal perceptions of others: giving to those who they view as generous, and exploiting/punishing those who they view as selfish. Likewise, if indirect positive and negative reciprocity play a role in structuring the game-play networks, then we expect individuals to act upon generalized perceptions of others: giving to those who are generally perceived by the community to be generous and exploiting/punishing those who are generally perceived by the community to be selfish.

Give this, the present study pays special attention to the ideas introduced in Bhui et al.[58], and investigates the following set of predictions:

P1 Behavior in network-structured economic games will be influenced by perceptions of social standing, with individual $j$:
  (a) giving to those they directly nominate as generous and failing to give to those they nominate as selfish,
  (b) not exploiting those they directly nominate as generous and exploiting those they nominate as selfish, and
  (c) not punishing those they directly nominate as generous and punishing those they nominate as selfish.

P2 Similarly, perceived standing will be influenced by cooperative behavior, with individual $j$'s perception of individual $k$ tracking how $k$ treats $j$ on an ongoing basis. Assuming that behavior in the economic games parallels or proxies for behavior in real-world contexts (i.e., that individual $j$ is more likely to give to individual $k$ in the allocation game if $j$ gives to, or shares with, $k$ in real-world contexts; see Pisor et al.[20] for evidence from Colombia), this leads to predictions that:
  (a) Individual $j$ will perceive individual $k$ to be generous if $k$ gave to $j$, and/or avoided exploiting/punishing $j$,
  (b) Individual $j$ will perceive individual $k$ to be selfish if $k$ exploited $j$, and/or avoided giving to $j$.

P3 After accounting for such dyadic perceptions of standing, generalized, community-wide reputations will influence economic game play, with:
  (a) individuals who are generally perceived to be generous being preferential targets of giving, and non-targets of exploitation or punishment, and
  (b) individuals who are generally perceived to be selfish being preferential targets of exploitation and punishment, and non-targets of giving.

To tease apart dyadic and generalized effects, we introduce a multiplex Bayesian generalization of the social relations model[14,65,66]. We apply the model to four multi-layer network datasets (see Fig. 1 and 2, and Table 1) collected in rural Colombia[20,21,67].

## Methods
### Ethnographic background
Due to increasing concerns about replicability in the social sciences[68,69], we repeated our study in four Colombian communities: a coastal community ($n = 186$ egos and 220 alters), a lowland community ($n = 154$ egos and 178 alters), a highland community ($n = 45$ egos and 53 alters), and an altiplano community ($n = 111$ egos and 160 alters). Data on age, gender, and ethnicity were not collected during the protocol for this particular study: however, overall, the participant pool in the long-form study database self-report as 58% female, 42% male, 44% Afrocolombian, 11% Emberá, and 45% Mestizo, with an average age of 39.8 years ($sd = 17.9$). Here, we refer to individuals who took part in the RICH games that were conducted in our study as 'egos' (i.e., the individuals who were making decisions in each game), and

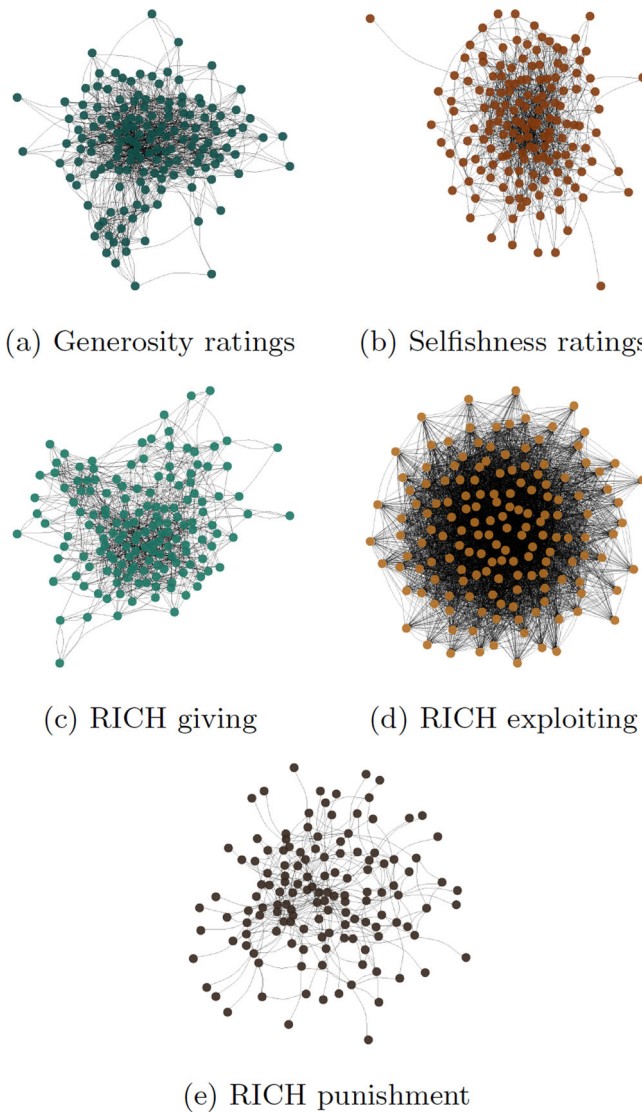

(a) Generosity ratings (b) Selfishness ratings

(c) RICH giving (d) RICH exploiting

(e) RICH punishment

**Fig. 1 | Digraphs of two dyadic peer-ratings networks. (a)** shows generosity ratings, and (**b**) selfishness ratings---and behaviors of three network-structured economic games---(**c**) shows the outcome network of the Rich giving game, (**d**) the outcome of the RICH exploitation game, and (**e**) the outcome of the RICH punishment game---from the lowland community in rural Colombia (see Figs. S1–4 for corresponding plots from the other communities). Each network layer is dense, impeding visual assessments of network structure. To gain a better understanding of how perceptions of generosity and selfishness structure behavior in the game networks, we use hive plots in Fig. 2.

refer to those who were targets of behaviors during the economics games as 'alters' (e.g., the set of individuals who were allocated resources, exploited, or punished).

In the coastal and lowland communities, the population is composed of a majority of Colombians of African descent, along with minorities of Mestizo and indigenous Emberá descent[70,71]. Both communities have been impacted by Colombia's internal conflicts, and violence from guerilla and paramilitary groups[72,73], and many individuals in each community are considered internally displaced persons within Colombia, having resettled after being forced from their natal communities. In the highland and altiplano communities, the population is composed of a majority of Colombians of Mestizo descent. Regardless of location or ethnic background, most individuals in these communities are living under rather austere socioeconomic circumstances. As such, social relationships are expected to play a critical role in buffering the challenges associated with long-term poverty and short-term fluctuations in resource security[20,74]. Beyond simple resource

considerations, social relationship are also important for adapting to a multitude of challenges impacting rural residents, including the problems of organizing cooperative labor, managing land-use, and providing childcare[15].

In terms of subsistence, the coastal community relies primarily on a mixture of fishing and local wage labor, along with limited levels of hunting, horticulture, and animal husbandry. The lowland community is located in the rain-forests of western Colombia, and relies primarily on a mixture of horticulture and local wage labor, but hunting, fishing, and animal husbandry are also practiced, as is small-scale gold panning. The highland community is located close to the lowland community and relies on small-scale agricultural production of coffee and sugarcane. The altiplano community is located close to the national capital, and residents primarily rely on wage labor, especially in companies focused on large-sale flower cultivation.

### Data collection

In each community, participants were invited to take part in the "RICH" network-structured economic games [see[20,63], for methodological details] and complete additional dyadic peer ratings on social standing measures (i.e., selfishness and generosity). Data were collected and coded using the `DieTryin` R package[75,76]. All participants provided informed consent to take part in the study, and understood that earnings would be paid in real money upon completion of the economic games. In each community, respondents were presented with a randomized photo array of all adult community members to/from whom they could allocate/take coins or tokens. In these RICH games, focal individuals can use their knowledge of the characteristics of alters to decide how to allocate money or reduction tokens, without allowing alters to know who gave (or did not give) coins or tokens to them[20].

In the RICH allocation game, respondents were: (i) presented with a small allotment of coins, (ii) told that they could keep as much of the allotment for themselves as they wanted by placing coins on their own photos, and (iii) told that they can make transfers to others in the community by placing coins on the photos of those individuals. In the RICH exploitation game, respondents were: (i) shown that a coin was pre-allocated to each community member, (ii) told that these coins would remain in the accounts of the others unless taken, and (iii) told that they could anonymously take from others in the community by removing the coins from the photograph roster, benefiting themselves at a cost to others. Finally, in the RICH costly reduction game, respondents were: (i) presented with a small allotment of coins, (ii) told that they can either keep these coins, or use them to purchase punishment tokens, and (iii) told that a punishment token will reduce the payout of a punished person with a 4 x multiplier.

The stakes per person for the RICH allocation game were set at 20,000 Colombian pesos at the lowland community, 25,000 Colombian pesos at the coastal community, and 30,000 Colombian pesos at the highland and altiplano communities (~5–8 USD). Individuals could allocate any number of 1000 peso coins to any cell in the photo array, including their own.

The stakes per person for the RICH exploitation game were set at 80,000 Colombian pesos at the altiplano community, 89,000 Colombian pesos at the lowland community, and 110,000 Colombian pesos at the coastal community (~20–25 USD). Individuals could take or leave the single 500 peso coin pre-allocated to each photo in the array. Due to a smaller sample size, stakes were set to 53,000 Colombian pesos at the highland community, where individuals could take or leave the single 1,000 peso coin pre-allocated to each photo in the array.

The stakes for the RICH costly reduction game were set at 15,000 Colombian pesos (~3 USD) in all four communities. Individuals could allocate any number of 1000 peso coins to the purchase of tokens, which could be then used to reduce the payout of any alter by 4000 pesos per token.

Finally, dyadic peer ratings for social standing/reputation were elicited by asking participants to place tokens on the photographs of community members who were especially generous (green tokens) or selfish (purple tokens). There was no minimum or maximum limit on the number of tokens that could be placed by each respondent, but most respondents used around 7 to 9 tokens per rating category.

**Fig. 2 | Directed ties observed in each community during the three network-structured games.** Here, each hive plot[115,116] represents a set of individuals ranked from bottom to top along each diagonal line in terms of their *reputation* measure: the sum total of nominations as *generous* minus the sum total of nominations as *selfish*. The ranking is smooth, but for visualization purposes we bin individuals into three separate reputation categories. This was done by coding individuals as having a `high' reputations when the count of their nominations for generosity exceeded their nominations for selfishness sufficiently, and vice versa for `low' reputation. Individuals viewed as more generous tend to give more, as is indicated by more lines flowing from higher up on the left-most axis. There is substantial variation in who receives allocations in each community---e.g., many poorer people, who could not afford to be as generous in the game, were still targets of giving. Exploitation is common in each community, and there is a clear signal that those with generous reputations preferentially exploit those with selfish reputations. Individuals of all reputational levels appear to punish at roughly equal rates, but people with selfish reputations are preferential targets of punishment.

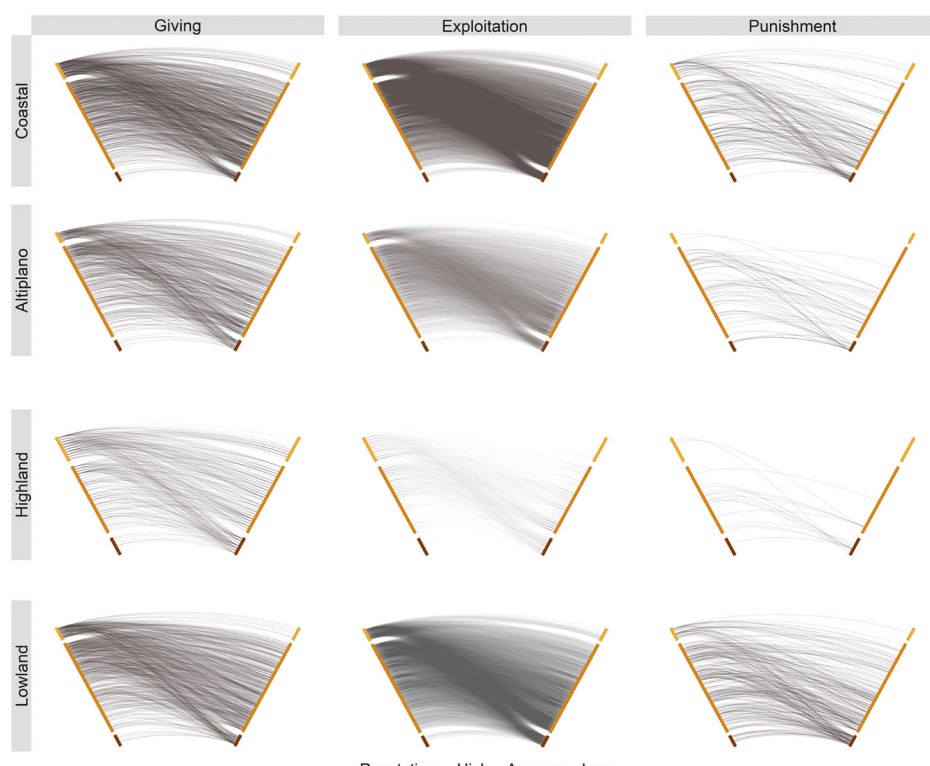

Reputation: — High — Average — Low

### Table 1 | Descriptive statistics of the network data by community

| Network | Vertices | Edges | Density | Avg. Dist. | Reciprocity | Transitivity |
|---|---|---|---|---|---|---|
| Coastal community | | | | | | |
| Generosity rating | 186 | 1732 | 0.05 | 3.027 | 0.197 | 0.21 |
| Selfishness rating | 186 | 1365 | 0.04 | 3.321 | 0.088 | 0.14 |
| RICH Giving | 186 | 1763 | 0.051 | 2.993 | 0.213 | 0.181 |
| RICH Exploitation | 186 | 20362 | 0.592 | 1.392 | 0.626 | 0.838 |
| RICH Punishment | 186 | 387 | 0.011 | 2.741 | 0.039 | 0.025 |
| Lowland community | | | | | | |
| Generosity rating | 154 | 1347 | 0.057 | 2.885 | 0.22 | 0.204 |
| Selfishness rating | 154 | 1072 | 0.045 | 3.12 | 0.097 | 0.16 |
| RICH Giving | 154 | 1271 | 0.054 | 3.114 | 0.195 | 0.194 |
| RICH Exploitation | 154 | 13607 | 0.577 | 1.412 | 0.589 | 0.846 |
| RICH Punishment | 154 | 480 | 0.02 | 3.718 | 0.048 | 0.083 |
| Highland community | | | | | | |
| Generous rating | 45 | 256 | 0.129 | 2.305 | 0.336 | 0.364 |
| Selfish rating | 45 | 115 | 0.058 | 2.815 | 0.139 | 0.155 |
| RICH Giving | 45 | 444 | 0.224 | 2.111 | 0.364 | 0.47 |
| RICH Exploitation | 45 | 810 | 0.409 | 1.559 | 0.427 | 0.682 |
| RICH Punishment | 45 | 62 | 0.031 | 1.261 | 0 | 0 |
| Altiplano community | | | | | | |
| Generous rating | 111 | 596 | 0.049 | 3.357 | 0.273 | 0.302 |
| Selfish rating | 111 | 319 | 0.026 | 3.794 | 0.182 | 0.13 |
| RICH Giving | 111 | 1020 | 0.084 | 2.873 | 0.314 | 0.302 |
| RICH Exploitation | 111 | 6690 | 0.548 | 1.43 | 0.549 | 0.823 |
| RICH Punishment | 111 | 178 | 0.015 | 2.587 | 0.119 | 0.019 |

Vertices indicate the number of individuals in the network. Edges are the number of directed ties. Density is the fraction of edges observed in the network over all possible edges. Average distance is the average number of edges between any two nodes in the network. Reciprocity is the probability that if individual *j* sends a tie to individual *k*, that individual *k* also sends a tie to individual *j*. Transitivity is the probability that adjacent nodes of a network are connected; if individual *j* is connected to individual *k*, and individual *k* is connected to individual *l*, transitivity measures the probability that individual *j* is also connected to individual *l*.

## Ethics and inclusion statement

All field protocols were approved by the Department of Human Behavior, Ecology, and Culture at the Max Planck Institute for Evolutionary Anthropology in Leipzig, Germany. Following local norms, informed consent was obtained from each respondent prior to data collection, and from the community leader or local community council, when appropriate. The data presented here are part of a longer-running, multi-year study of inequality, poverty, and social support networks, with relevance for both academic and applied research. Although research has been ongoing for >5 years, informed consent is re-obtained at each each wave of data collection. Because literacy is sometimes limited in the local population, informed consent is obtained verbally after providing participants with a verbal description (in Spanish) of the research process and explaining how data will be used (anonymously, for research purposes); in addition, participants are provided with a written consent document. This research project was conducted as part of a long-term collaboration with Afrocolombian researchers A.H.M. and D.H.M. who, along with C.T.R., collected data and managed field research.

## Analytical strategy

**The social relations model.** Our analysis of the multiplex structuring of network ties across the three RICH economic games and two social standing variables is based on the Social Relations Model (SRM)[14,65,66,77]. The standard SRM investigates generalized and dyadic reciprocity *within* a given network layer. The terms "generalized reciprocity" and "dyadic reciprocity" have technical definitions in the literature related to the SRM. Such definitions differ from the usage of similar words in other literatures. We provide precise definitions of these terms below when discussing the relevant parameters of our model. We use the term *generalized reciprocity* in a way that remains consistent with previous research that applies the social relations model[14,20,21,23]. Our usage should not be confused with previous uses of the term "generalized reciprocity" in behavioral ecology that refer to situations where a giver allocates resources if it has been given before [e.g., 'paying it forward'[78,79], or uses in economic anthropology that refer to situations where givers do not expect receivers to pay an equal amount of resources back in a pre-determined time period[80]. In this paper, we provide an extension of the SRM that investigates generalized and dyadic reciprocity both within *and between* network layers. This more general model structure allows us to investigate if—for example—individuals who are rated as generous are less likely to be exploited in the RICH taking game or punished in the RICH costly reduction game.

Assume that we have collected $M$ network layers of economic game and peer ratings data on a photograph roster in which a total of $J$ alters appear. Then, our outcome variable of interest, $G$, is an indicator variable describing if a coin or token was placed by individual $j$, on alter $k$, in network layer $m$, which we notate as: $G_{[j,k,m]} \in \{0, 1\}$.

We can model these outcomes jointly using a Bernoulli regression model:

$$G_{[j,k,m]} \sim \text{Bernoulli}\left(\text{logistic}\left(\theta_{[j,k,m]}\right)\right) \tag{1}$$

where the parameter $\theta_{[j,k,m]}$ gives the log-odds of a directed transfer from individual $j$ to individual $k$ in game $m$. To parameterize the model, we integrate sender, recipient, and dyadic random effects, along with any desired predictor variables following the standard SRM structure:

$$\theta_{[j,k,m]} = \eta_{[m]} + \alpha_{[j,m]} + \beta_{[k,m]} + \delta_{[j,k,m]} + \ldots \tag{2}$$

The intercept term in game $m$ is $\eta_{[m]}$. The first random effect term, $\alpha_{[j,m]}$, is the sender effect of individual $j$ in game $m$; this parameter measures the likeliness of individual $j$ directing a transfer towards any individuals other than the self in game $m$. The next random effect term, $\beta_{[k,m]}$, is the recipient effect of individual $k$ in game $m$; this parameter measures the likeliness of

individual $k$ receiving a transfer from any individual in the community in game $m$. The last random effect term, $\delta_{[j,k,m]}$, is a dyad-level random effect in game $m$; this parameter measures the likeliness of individual $j$ directing a transfer to individual $k$ in game $m$. The inclusion of random effects at the level of the binary outcome variable results in an underidentified model in non-Bayesian approaches[81]. Our Bayesian approach relies on priors to identify the model; the qualitative inferences about dyadic correlations are plausibly robust to such identifiability concerns. Analogous models, including the p2 model[82,83] and the data augmentation version of the probit SRM[84,85], can be fit to single-layer networks. Multiplex extensions of these models are conceptually possible, but have not yet been advanced in the literature. If desired, the ellipse in Eq. (2) can be replaced with a linear model for the effects of characteristics of the sender, receiver, or dyad. For example, controls for wealth, $W$, and relatedness, $R$, could be included by adding: $\lambda_{[1,m]}W_{[j]} + \lambda_{[2,m]}W_{[k]} + \lambda_{[3,m]}R_{[j,k]}$, if such data were available.

**A multi-layer generalization.** To complete the SRM structure, it is typical to use bivariate normal distributions to estimate both *generalized reciprocity* (i.e., the correlation between sender effects, $\alpha$, and recipient effects, $\beta$) and *dyadic reciprocity* (i.e., the correlation between flows from $j$ to $k$, $\delta_{[j,k]}$, and flows from $k$ to $j$, $\delta_{[k,j]}$). The standard SRM approach, however, is insufficient for our multiplex data, as there are important additional correlations that we need to measure. For example, the correlation of the $\alpha$ parameters across network types will indicate how likely an individual who gives a lot in the allocation game is to also punish a lot in the costly reduction game.

The sub-model for the generalized reciprocity terms can be extended simply by concatenating the sender and receiver effects for each layer into a single vector, and using a standard multivariate normal model:

$$\begin{pmatrix} \alpha[j,1] \\ \ldots \\ \alpha[j,M] \\ \beta[j,1] \\ \ldots \\ \beta[j,M] \end{pmatrix} \sim \text{M.V. Normal}(Z, \Sigma) \tag{3}$$

where $Z$ is a vector of zeroes and $\Sigma$ is a $2M \times 2M$ covariance matrix. Computationally [see:[86]], it is much more efficient to implement this model by instead defining:

$$\begin{pmatrix} \alpha_{[j,1]} \\ \ldots \\ \alpha_{[j,M]} \\ \beta_{[j,1]} \\ \ldots \\ \beta_{[j,M]} \end{pmatrix} = \sigma_\circ \left( L * \begin{pmatrix} \hat{\alpha}_{[j,1]} \\ \ldots \\ \hat{\alpha}_{[j,M]} \\ \hat{\beta}_{[j,1]} \\ \ldots \\ \hat{\beta}_{[j,M]} \end{pmatrix} \right) \tag{4}$$

where $\sigma$ is a vector of standard deviations, $L$ is a Cholesky factor of a $2M \times 2M$ correlation matrix, the symbol $*$ denotes the standard matrix product, and the symbol $\circ$ denotes the Hadamard, or element-wise, product. When all of the raw random effects are given unit normal priors:

$$\hat{\alpha}_{[j,m]} \sim \text{Normal}(0, 1) \tag{5}$$

$$\hat{\beta}_{[j,m]} \sim \text{Normal}(0, 1) \tag{6}$$

then Eq. (4) is equivalent to Eq. (3)[86]. The sub-model is completed by putting weak priors on the standard deviations and the Cholesky factor:

$$\sigma \sim \text{Exponential}(2.5) \tag{7}$$

$$L \sim \text{LKJ Cholesky}\,(2.5) \tag{8}$$

We can use a similar approach for the dyad-level random effects. In the standard SRM, the dyadic reciprocity coefficient measures the extent to which inter-personal flows are bidirectional—i.e., it tests if when focal $j$ gives more to alter $k$, that $k$ also gives more to $j$. Our generalized model also tests for cross-layer dyadic reciprocity—e.g., it tests if when individual $j$ gives more to individual $k$, that individual $k$ is less likely to take from individual $j$. It also tests if behaviors are correlated within individual $j$—e.g., it tests if when individual $j$ gives more to individual $k$ that individual $j$ also engages in less costly punishment of individual $k$.

The definition of the dyadic reciprocity sub-model follows a similar form to the generalized reciprocity sub-model; however, there are some additional constraints on the standard deviation and correlation parameters that we must account for in this case. Nevertheless, the first step is the same in both models. The dyadic random effects are concatenated across network layers and modeled as before:

$$\begin{pmatrix} \delta_{[j,k,1]} \\ \dots \\ \delta_{[j,k,M]} \\ \delta_{[k,j,1]} \\ \dots \\ \delta_{[k,j,M]} \end{pmatrix} = \varsigma_\circ \left( \Gamma * \begin{pmatrix} \hat{\delta}_{[j,k,1]} \\ \dots \\ \hat{\delta}_{[j,k,M]} \\ \hat{\delta}_{[k,j,1]} \\ \dots \\ \hat{\delta}_{[k,j,M]} \end{pmatrix} \right) \tag{9}$$

where $\varsigma$ is a vector of standard deviations, and $\Gamma$ is a Cholesky factor of a $2M \times 2M$ correlation matrix. As before, the raw random effects have unit normal priors:

$$\hat{\delta}_{[j,k,m]} \sim \text{Normal}\,(0, 1) \tag{10}$$

and weak priors are placed on the standard deviations and the correlation matrix Cholesky factor:

$$\varsigma \sim \text{Exponential}\,(2.5) \tag{11}$$

$$\Gamma \sim \text{LKJ Cholesky}\,(2.5) \tag{12}$$

At this point, we can address the special constraints that are needed in the dyadic reciprocity sub-model. First, for a generalized SRM of the type we present here, $\varsigma$ will be a vector of length $2M$, where $M$ is the number of network layers. The variance of the $\delta_{[j,k,m]}$ parameters must match the variance of the $\delta_{[k,j,m]}$ parameters, as these are realization from the same distribution. To impose this constraint, we fix $\varsigma$ to use only $M$ free parameters, and define the other $K$ parameters by writing:

$$\varsigma_{[m]} = \varsigma_{[m+M]} \tag{13}$$

Similarly, the correlation matrix in this model requires a special symmetry property. For example, the parameter that measures the correlation between the propensity of individual $j$ to give coins to $k$ and pay coins to reduce $k$, should be equal to the parameter that measures the correlation between the propensity of individual $k$ to give coins to $j$ and pay coins to reduce $j$, as the labels $j$ and $k$ are arbitrary. Accounting for all of these similar constraints, one finds that the dyadic correlation matrix, $\rho$, must be of the special block structure:

$$\rho = \begin{pmatrix} C & B \\ B & C \end{pmatrix} \tag{14}$$

where $C$ is a valid $M \times M$ correlation matrix, the elements of $B \in (-1, 1)$, and $B$ is equal to its own transpose ($B = B^T$).

Unfortunately, there is no known method for generating correlation matrices with this property that guarantees the resultant matrix will be

positive definite. Nevertheless, we can use standard methods to generate positive definite correlation matrices and then use priors to force $\rho$ arbitrarily close to the desired symmetry condition.

First, we note that $\rho$ can be generated by multiplying $\Gamma$ by its own transpose:

$$\rho = \Gamma\Gamma^T \tag{15}$$

and then we note that the posterior distribution of $\rho$ can be forced to take the form given in Eq. (14) by using priors to penalize two particular $\ell^2$ norms. Specifically, for $m \in \{1, \dots, M-1\}$ and $n \in \{m+1, \dots, M\}$, we model:

$$\| \rho_{[m+M,n+M]} - \rho_{[m,n]} \| \sim \text{Normal}\,(0, \epsilon) \tag{16}$$

to constrain the $C$ blocks in Eq. (14), and:

$$\| \rho_{[m,n+M]} - \rho_{[n,m+M]} \| \sim \text{Normal}\,(0, \epsilon) \tag{17}$$

to constrain the $B$ blocks in Eq. (14). In the limit, as $\epsilon \to 0$, the posterior of $\rho$ takes on the form given in Eq. (14). In practice, we set $\epsilon$ equal to a small constant—e.g., 0.1.

### Software
Data analysis was conducted in R[87]. Statistical models were coded in Stan and fit using the `rstan` package[88]. We diagnosed model fits and Markov Chain Monte Carlo performance using trace plots, $\hat{R}$, and reported effective samples. All diagnostics indicate good model fit.

### Reporting summary
Further information on research design is available in the Nature Portfolio Reporting Summary linked to this article.

## Results
Closely following the expectations of formal models, we find that cooperation, exploitation, and punishment are all tightly linked to perceptions of social standing—at both dyadic and generalized levels. We begin by presenting our results on the dyadic structure of behavior—e.g., how individual $j$'s behavior towards individual $k$ is associated with $j$'s perception of the social standing of $k$. We then present results on individual-level structure in behavior—e.g., we test if individuals with high generosity scores (based on community-wide response patterns) are more likely to be the targets of cooperative donations, above and beyond what is explained by dyadic effects.

### Dyadic structure
Behavior in each game was characterized by dyadic reciprocity, both within and between network layers. In the following sections, we outline our tests of predictions P1 and P2. We present estimates—posterior means and 89% credible intervals—from the coastal community using the symbol $\rho_c$, the lowland community using the symbol $\rho_l$, the highland community using the symbol $\rho_h$, and from the altiplano community using the symbol $\rho_a$. Figure 3 plots a matrix of dyadic reciprocity coefficients from the lowland community. Figure 3b and c plot the full suite of estimates from all four communities, and demonstrate a tight replication of most findings across locations.

**Positive and negative direct reciprocity structure economic behavior and perceptions of social standing.** Within layers, we find that individuals reciprocally give to one another in the allocation game ($\rho_l = 0.61$, CI: 0.52, 0.68; $\rho_c = 0.70$, CI: 0.63, 0.77; $\rho_h = 0.63$, CI: 0.48, 0.77; $\rho_a = 0.72$, CI: 0.65, 0.79), take from one another in the exploitation game ($\rho_l = 0.21$, CI: 0.15, 0.27; $\rho_c = 0.53$, CI: 0.48, 0.57; $\rho_h = 0.53$, CI: 0.37, 0.70; $\rho_a = 0.36$, CI: 0.29, 0.43), and punish one another in the costly reduction game ($\rho_l = 0.40$, CI: 0.21, 0.58; $\rho_c = 0.46$, CI: 0.25, 0.65; $\rho_a = 0.56$, CI: 0.36, 0.74)—only in the highland site, where punishment was exceedingly rare do we observe a lack of dyadic reciprocity ($\rho_h = -0.01$, CI: -0.48, 0.48).

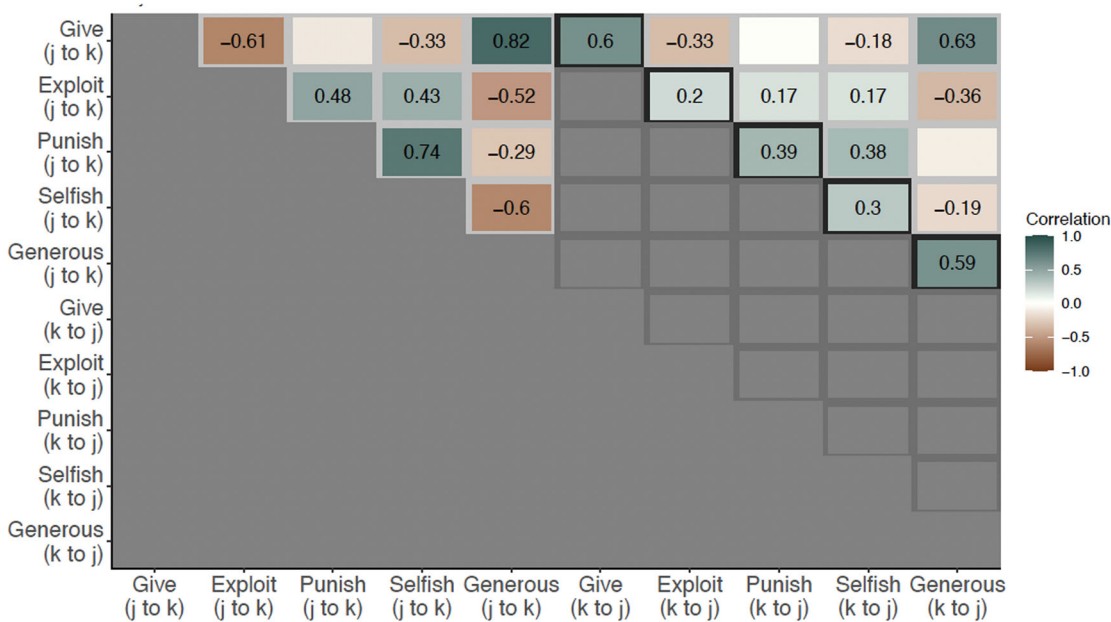

(a) Dyadic Correlations

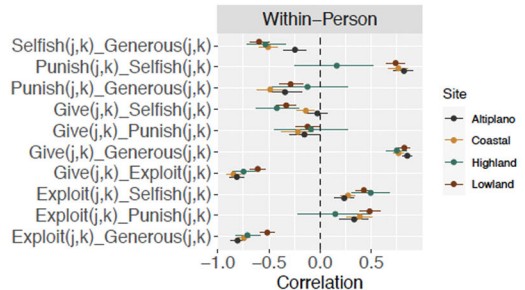

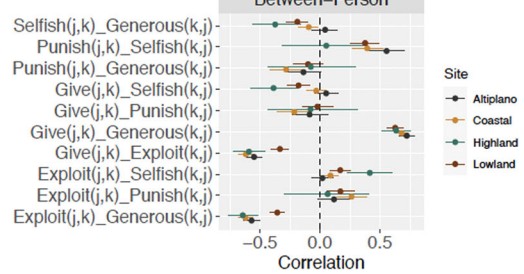

(b) Within-person, between layer correlations (89% CI)

(c) Between-person, between layer correlations (89% CI)

**Fig. 3 | Dyad-level correlations in random effects.** Frame (**a**) plots posterior mean values of all dyadic correlation parameters from the lowland community organized in matrix form (See Figs S5, S7, S9 and S11 in the supplementary Materials for the plots for all communities). For example, we see that if individual *j* gave to individual *k* in the giving game, then individual *j* is reliably less likely to exploit individual *k* in the exploitation game ($\rho = -0.61$). The left-most (gray) triangle of estimates gives within-person, between-layer correlations in dyadic random effects. The right-most (gray) triangle of estimates gives between-person, between-layer correlations in dyadic random effects. The diagonal set of estimates (black) gives within-layer dyadic reciprocity. Only reliable correlations are shown. Frame (**b**) shows the within-person, between-layer correlations as posterior means and 89% credible intervals (*CI*). Frame (**c**) Between-person, between-layer correlations as posterior means and 89% credible intervals. We note that social standing as *generous* is reliably associated with increased probability of receiving coins in the giving game, decreased probability of being exploited in the taking game, and decreased probability of being reduced in the costly punishment game. Likewise, social standing as *selfish* is reliably associated with decreased probability of receiving coins in the giving game, increased probability of being exploited in the taking game, and increased probability of being reduced in the costly punishment game.

Moreover, individuals mutually view one another as being either generous ($\rho_l = 0.30$, CI: 0.21, 0.40; $\rho_c = 0.31$, CI: 0.22, 0.40; $\rho_h = 0.39$, CI: 0.08, 0.68; $\rho_a = 0.57$, CI: 0.44, 0.69) or selfish ($\rho_l = 0.59$, CI: 0.52, 0.66; $\rho_c = 0.60$, CI: 0.55, 0.66; $\rho_h = 0.60$, CI: 0.44, 0.75; $\rho_a = 0.66$, CI: 0.58, 0.73).

**Individuals allocate resources to those in good standing.** Turning now to the correlations between network layers, we find support for prediction P1(a); participants behave more cooperatively towards those they perceive as being in good social standing. That is, in both communities, participants are more likely to give to individuals whom they rate as generous ($\rho_l = 0.82$, CI: 0.76, 0.88; $\rho_c = 0.77$, CI: 0.72, 0.82; $\rho_h = 0.75$, CI: 0.64, 0.85; $\rho_a = 0.85$, CI: 0.80, 0.90) and are less likely to give to individuals whom they rate as selfish ($\rho_l = -0.33$, CI: -0.42, -0.23; $\rho_c = -0.14$, CI: -0.22, -0.06; $\rho_h = -0.42$, CI: -0.63, -0.21)—this last effect is not reliably negative at the altiplano site, however, ($\rho_a = -0.03$, CI: -0.13, 0.07).

**Individuals exploit and punish those in bad standing.** Following prediction P1(b), we find that participants are less likely to exploit individuals that they rate as generous ($\rho_l = -0.53$, CI: -0.59, -0.46; $\rho_c = -0.74$, CI: -0.80, -0.68; $\rho_h = -0.71$, CI: -0.83, -0.58; $\rho_a = -0.81$, CI: -0.87, -0.73) and are more likely to exploit individuals whom they rate as selfish ($\rho_l = 0.43$, CI: 0.36, 0.51; $\rho_c = 0.27$, CI: 0.20, 0.33; $\rho_h = 0.49$, CI: 0.31, 0.67; $\rho_a = 0.24$, CI: 0.14, 0.33). Likewise, following prediction P1(c), we find that participants are less likely to punish individuals whom they rate as generous ($\rho_l = -0.28$, CI: -0.39, -0.16; $\rho_c = -0.51$, CI: -0.64, -0.37; $\rho_h = -0.12$, CI: -0.47, 0.27; $\rho_a = -0.34$, CI: -0.49, -0.18) and are more likely to punish individuals whom they rate as selfish ($\rho_l = 0.73$, CI: 0.64, 0.82; $\rho_c = 0.77$, CI: 0.67, 0.85; $\rho_h = 0.16$, CI: -0.25, 0.52; $\rho_a = 0.82$, CI: 0.72, 0.90). Note, however, that although the direction of effects at the highland site are consistent with other sites, the credible intervals are much wider (and do not exclude zero) since punishment was rare there, and the sample size was smaller.

**Individuals may gain good standing through cooperation.** Following predictions P2(a), we find that cooperative behavior is important for establishing and maintaining good standing with other individuals. Specifically, we show that if individual $j$ gives to individual $k$ in the game, then $j$ is substantially more likely to be rated as generous by individual $k$ ($\rho_l = 0.63$, CI: 0.56, 0.70; $\rho_c = 0.69$, CI: 0.63, 0.74; $\rho_h = 0.63$, CI: 0.52, 0.75; $\rho_a = 0.72$, CI: 0.66, 0.79). It is important to note that when individual $k$ is rating individual $j$ as selfish or generous, $k$ has no information about individual $j$'s game-play behavior. Thus, the fact that such a high correlation between $j$'s behavior and $k$ perception of $j$ emerges in our network data suggests that behavior in the economic games is accurately reflecting broader, ecologically valid patterns of on-going behavior in the real world.

Paralleling the above results, if individual $j$ exploits individual $k$, then $j$ is less likely to be rated as generous by individual $k$ ($\rho_l = -0.36$, CI: -0.42, -0.30; $\rho_c = -0.62$, CI: -0.67, -0.57; $\rho_h = -0.64$, CI: -0.76, -0.52; $\rho_a = -0.57$, CI: -0.64, -0.50). The same directed effect is observed in the case of punishment, but it is weaker in magnitude and less reliable: if individual $j$ punishes individual $k$, then $j$ is less likely to be rated as generous by individual $k$ ($\rho_l = -0.11$, CI: -0.23, 0.01; $\rho_c = -0.26$, CI: -0.41, -0.12; $\rho_h = -0.08$, CI: -0.44, 0.29; $\rho_a = -0.14$, CI: -0.28, 0.01).

**Individuals may gain bad standing through exploitation and punishment.** Following predictions P2(b), we find that cooperative behavior appears important for preventing the acquisition of selfish standing, while exploitation and punishment behavior is associated with being perceived of as selfish. Specifically, we show that if individual $j$ gives to individual $k$, then $j$ is less likely to be considered as selfish by individual $k$ at the lowland and highland sites ($\rho_l = -0.18$, CI: -0.27, -0.08; $\rho_h = -0.39$, CI: -0.58, -0.19). However, at the coastal and altiplano sites, effects are not reliably non-zero ($\rho_c = -0.03$, CI: -0.11, 0.04; $\rho_a = 0.05$, CI: -0.05, 0.16). Similarly, if individual $j$ exploits individual $k$, then $j$ is more likely to be considered as selfish by individual $k$ at three of the four sites ($\rho_l = 0.18$, CI: 0.07, 0.29; $\rho_c = 0.27$, CI: 0.14, 0.40; $\rho_h = 0.06$, CI: -0.30, 0.41; $\rho_a = 0.12$, CI: -0.02, 0.24). This same pattern holds with respect to punishment; if individual $j$ punishes individual $k$, then $j$ is more likely to be considered as selfish by individual $k$ at three of four sites ($\rho_l = 0.17$, CI: 0.08, 0.26; $\rho_c = 0.34$, CI: 0.22, 0.47; $\rho_h = 0.06$, CI: -0.29, 0.41; $\rho_a = 0.13$, CI: 0.00, 0.26).

**Individual-level network structure**
Having outlined the dyadic structure in game-play, we now turn to exploring individual-level variation. Our generalization of the social relations model permits estimation of random effects that govern individual-level propensities to send and receive network ties in each outcome layer, after accounting for the dyadic effects outlined above. Some individuals, for example, may be more likely to give to others, in general, and this would be reflected in elevated "sender effects" in the RICH giving game. Likewise, some individuals may be more likely to be given to by others, in general, and this would be reflected in elevated "receiver effects" in the RICH giving game. The correlations between these individual-level random effects vectors are often termed *generalized reciprocity* in the SRM literature. Cross-layer generalized reciprocity estimates may be used to measure—for example—if people who are widely rated as generous by many in the community are more likely to receive coins from others, above and beyond what would be expected from dyadic effects alone. Figure 4 plots a matrix of generalized reciprocity coefficients from the lowland community. Figure. 4b and c plot the full suite of estimates from all four communities.

**Reputations for generosity are associated with receipt of cooperative transfers from others, and elusion of exploitation.** Following prediction P3(a), we find that—at a generalized level—individuals with generous reputations are more likely to receive resources in the giving game ($\rho_l = 0.66$, CI: 0.54, 0.77; $\rho_c = 0.59$, CI: 0.48, 0.69; $\rho_h = 0.50$, CI: 0.28, 0.69; $\rho_a = 0.52$, CI: 0.36, 0.65) and are less likely to be exploited in the

taking game ($\rho_l = -0.21$, CI: -0.35, -0.04; $\rho_c = -0.42$, CI: -0.53, -0.31; $\rho_h = -0.52$, CI: -0.71, -0.31; $\rho_a = -0.43$, CI: -0.57, -0.28).

Contrary to our expectations, individuals with generous reputations in the lowland, highland, and altiplano communities are not less likely to be punished in the costly reduction game ($\rho_l = 0.05$, CI: -0.13, 0.23; $\rho_h = -0.13$, CI: -0.53, 0.32; $\rho_a = 0.13$, CI: -0.17, 0.41); in the coastal community, such individuals are actually *more* likely to be punished ($\rho_c = 0.23$, CI: 0.05, 0.41).

**Individuals punish those with a reputation for being selfish.** Following prediction P3(b), we find that, at a generalized level, individuals with selfish reputations are more likely to be punished in the costly reduction game ($\rho_l = 0.84$, CI: 0.74, 0.93; $\rho_c = 0.73$, CI: 0.62, 0.84; $\rho_h = 0.12$, CI: -0.34, 0.52; $\rho_a = 0.57$, CI: 0.30, 0.78). However, in contrast to our predictions, we did not observe that individuals with selfish reputations are more likely to be taken from during the exploitation game ($\rho_l = 0.04$, CI: -0.10, 0.18; $\rho_c = -0.09$, CI: -0.21, 0.04; $\rho_a = -0.16$, CI: -0.34, 0.01), except at the highland site ($\rho_h = 0.36$, CI: 0.11, 0.59).

## Discussion
Theoretical models have highlighted the potential importance of punishment[89], positive indirect reciprocity[49], and—more recently—negative indirect reciprocity[58], for generating and stabilizing large-scale cooperation. Here, we report findings from network-structured economic games in four rural Colombian communities. Our results provide evidence that is consistent with core predictions about the importance of both positive and negative, direct and indirect, reciprocity for the maintenance of cooperation in humans. We find that individuals condition cooperative, exploitative, and punishment behavior on dyadic inter-personal standing: cooperating with those they believe to be generous, and exploiting and punishing those they consider selfish. These dyadic findings scale to a more generalized, community level, where reputations for being generous are associated with receipt of allocations, and reputations for being selfish are associated with receipt of punishment, further showing how shared social perceptions guide inter-personal behavior.

### Generosity and cooperation
The importance of standing and reputation for the maintenance of cooperative networks has been a topic of interest for decades[48,49,90,91]. Social monitoring and gossip can lead to discriminate partner choice, with individuals forming cooperative ties with those in good standing, as such group members can be better trusted as reciprocal partners [e.g.,[24,92–95]]. Our results support these models, showing that individuals treat others differently as a function of whether those others are perceived to be generous or selfish. Alongside this, we show that dyadic perceptions of generosity and selfishness (individual $j$ rating $k$) seemed to track behavior in the RICH games (individual $k$ transferring to $j$); in other words, dyadic perceptions of generosity and selfishness tended to accurately track dyadic resource flows in the game, even though economic behavior *in the game* was private (speaking to the ecological validity of the games).

In line with predictions from recent models [e.g.,[58]], individuals were not only more likely to perform costly acts of cooperation (i.e., allocate resources) towards those whom they believe are generous, but were also much less likely to exploit or punish such individuals. Finally, we observed evidence of reputation-based partner choice[56,96,97], with individuals directing cooperative behavior towards those who were considered more generous by the community at large. Taken together, our results highlight a distinct behavioral profile for individuals in good standing: they are highly cooperative and allocentric.

### Exploitation and punishment
The present findings also shed light on the network structure of exploitation and punishment behavior, and provide evidence of their roles in sustaining cooperative networks. Given the associations that we observed, it appears that selfish or anti-social behavior can be regulated through both

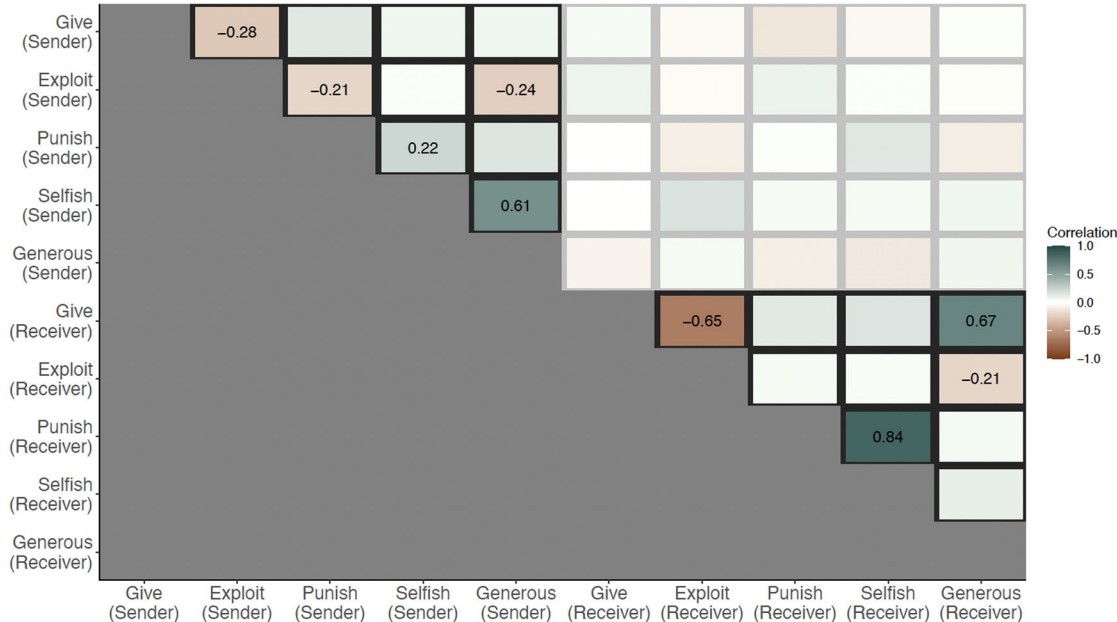

(a) Correlations in individual-level random effects.

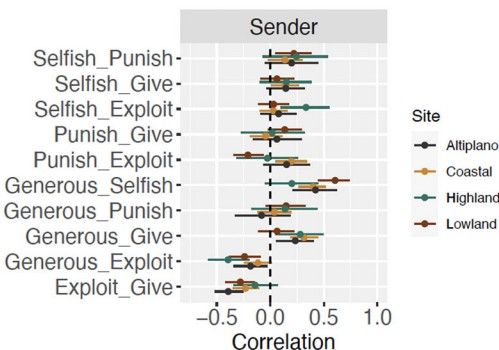

(b) Generalized correlations in sender effects as posterior means and 89% credible intervals.

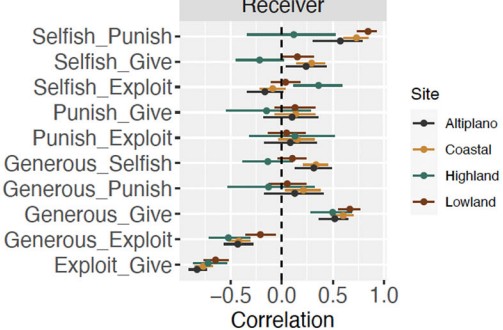

(c) Generalized correlations in receiver effects as posterior means and 89% credible intervals.

**Fig. 4 | Generalized correlations in individual-level sender and receiver effects (See Figs. S6, S8, S10 and S12 in the Supplementary Materials for the plots for all communities).** Frame (**a**) plots posterior mean values of all generalized reciprocity correlation parameters from the lowland community organized in matrix form. For example, we see that if an individual was on average a target of giving in the allocation game---see row labeled *Give (Receiver)*---then that same individual is also likely to be a target of high generosity ratings---see row labeled *Generous (Receiver)*---on average ($\rho_l = 0.66$). The upper-most (black) triangle of estimates gives generalized correlations in sender effects---e.g., if an individual tends to give to others overall, how likely are they to exploit others overall? The lower-most

(black) triangle of estimates gives generalized correlations in receiver effects---e.g., if an individual tends to be given to by others overall, how likely are they to be exploited by others overall? The correlations between sender and receiver effects appear in the gray square region, but were not reliably non-zero in these models. Frames (**b**) and (**c**) provide posterior credible intervals for the effects outlined in frame (**a**). We note that generalized social standing as *generous* is reliably associated with increased probability of receiving coins in the giving game and decreased probability of being exploited in the taking game. Similarly, generalized social standing as *selfish* is reliably associated with an increased probability of being reduced in the costly punishment game.

exploitation and costly punishment of non-cooperative (or even exploitative) individuals[98,99]. We also find evidence of reciprocal punishment[100].

Interestingly, we do not find a consistent behavioral profile for those who punish: the nexus between punishment and exploitation differed appreciably between communities. Although individuals who punished others more broadly were more likely to consider many other individuals as having a bad reputation (in all communities), the relationships between sender effects for punishing, giving, and exploiting were not consistent across communities. In the lowland community: (i) punishment and exploitation rates were negatively correlated, and (ii) individuals who gave more to others also paid more to punish others. In this case, it looks like pro-social individuals are more generous in the giving game, less-likely to indiscriminately exploit others, and more likely to pay to punish

(presumably anti-social) individuals. However, in the other communities, pro-social individuals (those who were more generous in the giving game) were less-likely to indiscriminately exploit others, but they were not more likely to pay to punish other (presumably anti-social) individuals.

Additionally, the decider profile of those who punish differs by community: coastal and altiplano punishers are exploitative and not generous, whereas lowland punishers are generous and not exploitative. Such a pattern might emerge if punishers at the coastal community are more needy and view taking from wealthier others—just like punishing wealthier others—as a method of social leveling. Such conditions may produce potentially 'negative' associations between punishment and reputation, speaking to previous research that has observed punishers as being perceived more negatively[96,101]. If punishers in the more integrated lowland community are

less needy and/or view taking from others as morally wrong (e.g., due to a higher presence of Catholic norms), they may be likely to behave like classic norm enforcers (punishing wrongdoers, but refraining from using punishment as a method of leveling). Indeed, some research has shown that such punishers may sometimes be held in esteem[102,103]. In sum, these results provide some supportive evidence for the co-occurrence of prosociality and punishment in the lowland community, but not in the other communities. Future work that links individual-level variation in game-play behavior to individual-level metrics of need, religiosity, and social status is needed to more thoroughly explore the topic.

### Negative indirect reciprocity
Formal theoretical models have proposed that preferential exploitation of individuals with selfish reputations can create an incentive structure that facilitates the emergence and maintenance of cooperation[58]. To the best of our knowledge, there are currently no empirical results that speak to the interaction between exploitation and reputation. Our results on negative indirect reciprocity here are mixed. At a dyadic level, individuals did not allocate resources to—and also paid to punish—those who exploited them. Yet, at the community level, individuals who exploited many others were not more or less likely to be punished—perhaps because exploitation is conditional, directed, and often covert. Similarly, individuals generally perceived by community members as having selfish reputations were more likely to be punished, but were not more or less likely to be exploited.

### Ecological validity
The current study provides some methodological advances for the empirical study of cooperation. True experimental paradigms have been instrumental in identifying the causal effects of certain mechanisms (e.g., indirect reciprocity) on cooperation rates [e.g.,[104–107]]. Whether such patterns generalize to real-world settings, however, is a matter of debate, as the majority of psychological research has been conducted using homogeneous samples [e.g., undergraduate students,[108]] in hypothetical or abstracted social environments [i.e., there is possibly limited ecological validity;[109,110]]. Our approach is not experimental (i.e., we do not manipulate an independent variable), but we show that perceptions of social standing are tightly correlated with behavioral game-play. We interpret the behavioral patterns observed during game-play as if they capture causal associations between interpersonal sentiments and ongoing social relationships, as many people described their game-play behavior in such a way—e.g., while speaking aloud during the games and narrating their motivations. Although RICH economic games are simple, and the stakes are relatively low (but substantial in the context of the local economies), we qualitatively observed that respondents put a lot of thought into their behavior—often agonizing over decisions. This is especially true for the exploitation game, where respondents typically spent several minutes considering who it was acceptable to take coins from for their own benefit: many factors were commented on verbally during the games—e.g., people would state that they took from those with substantial material wealth, especially if those people were considered stingy, but avoided taking from those most in-need (see Gervais[63] for similar findings in Fiji). For more details on ecological validity, and comparisons of RICH game-play behavior with self-reports of food/money transfers, see Pisor et al.[20].

### Unexpected associations
The analysis of between-layer generalized reciprocity revealed a number of unexpected associations. First, individuals who were widely rated as generous were also more likely to be punished in the lowland community. This may be considered a form of antisocial punishment—which has been observed during public goods games, to a greater or lesser extent, across societies[111–114]. Alternatively, it might reflect an empirical phenomenon where some wealthy individuals are viewed as generous by a subset of the community, but are seen as selfish by others. Second, there was small, but reliable, tendency for individuals who were widely rated as selfish to be

preferential targets of giving, as well as punishment. We believe that this effect is likely driven by economic need—the poorest individuals in each community are likely to be seen as selfish, as they lack the resources needed to gain a generous reputation. Certain individuals may give to these community members due to their need. Finally, in the coastal and altiplano communities, individuals who are widely rated as generous are also widely rated as selfish. This indicates a lack of consensus in those communities, with different alters rating the same egos in very different ways. In sum, these complicated findings highlight the importance of not relying on fully-aggregated reputation measures, and instead considering the dyadic structure of social standing and inter-personal behavior.

### Limitations
While the present research provides support for the importance of positive and negative, direct and indirect, reciprocity for sustaining cooperation, the study is correlational. Inclusion of experimental treatments and interventions in RICH network-structured economic games is a promising avenue for future research. Doing this would allow future studies to tease apart the causal effects of certain mechanisms that are driving cooperative behavior in real-world settings. Longitudinal research building upon the present findings would also be instrumental for exploring the dynamics of positively and negatively valanced dyadic behavior. Alongside this, the current research solely focuses on two core measures of reputation that are believed to most directly impact cooperative behavior. An option for future research is to incorporate other forms of reputation, for example, distinct types of status[26,55], and different interpersonal sentiments[33], to examine how these socio-cognitive features influence networks of cooperation, exploitation, and punishment in human communities.

### Conclusions
In conclusion, standing and reputation have been crucial in molding human cooperative psychology, and continue to shape how and why humans are able to form and maintain large networks of cooperative relationships. Our data suggest that individuals condition their cooperative behaviors toward those in good standing, and that those in bad standing are targets of exploitation and punishment. These findings support theories about the roles of both direct and indirect reciprocity in the evolution of cooperation in a setting that is reflective of real-world behavior.

### Data availability
The data presented in this manuscript are collected as part of a wider, longitudinal field study on wealth, demography, and social networks. All data for diagnostics and analysis reproduction are openly available on GitHub at: www.github.com/ctross/cross_layer_correlations.

### Code availability
The computational code used to produce any results, tables or figures presented in the manuscript are openly available on GitHub at: www.github.com/ctross/cross_layer_correlations.

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

## Acknowledgements

We thank all of the respondents who took the time to collaborate with us, Jonas Platzek for assistance with data entry, and Ronny Barr for assistance with preparation of the photograph rosters. Bret Beheim provided useful R functions to streamline the LaTeXworkflow. The research project benefited from outstanding secretarial support from the Department of Human, Behavior, Ecology, and Culture at the Max Planck Institute for Evolutionary Anthropology. This research was funded by the Department of Human, Behavior, Ecology, and Cultural at the Max Planck Institute for Evolutionary Anthropology.

## Author contributions

Daniel Redhead designed the study, reviewed the literature and wrote the manuscript. Matthew Gervais developed the RICH games protocols and edited the manuscript. Kotrina Kajokaite and Jeremy Koster developed an analogous multiplex SRM model for binomial outcomes and edited the manuscript. Arlenys Hurtado Manyoma and Danier Hurtado Manyoma collected the data. Richard McElreath funded the research and edited the manuscript. Cody T. Ross designed the study, designed the empirical research protocols, collected the data, coded the statistical models, and wrote the manuscript.

## Funding

## Competing interests

All authors declare no competing interests.
