## [Peer Review File · Communications Psychology]

20th Jul 23

Dear Dr Redhead,

Thank you for your patience during the peer-review process and apologies for the delay in providing you with a decision. Your manuscript titled "Reputation, exploitation, and the maintenance of cooperation: Evidence of direct and indirect reciprocity in network-structured economic games" has now been seen by 3 reviewers, and I include their comments at the end of this message. They find your work of interest, but raised some important points. We are interested in the possibility of publishing your study in Communications Psychology, but would like to consider your responses to these concerns and assess a revised manuscript before we make a final decision on publication.

We therefore invite you to revise and resubmit your manuscript, along with a point-by-point response to the reviewers. Please highlight all changes in the manuscript text file.

All reviewers find your manuscript of interest and overall well conducted. They provide some valuable suggestions to improve the clarity of your conceptual framing, methods and results, which will enable you to strengthen your manuscript.

The reviewers highlight some aspects of the results where the data cannot disambiguate between competing explanations. We ask you to include mention of these remaining ambiguities in a section titled "Limitations" in the Discussion, avoiding any speculation.

Reviewer #3 asks you to clarify your predictions. Please note that you can provide further justification for your predictions in the introduction section and rephrase the language of your predictions, however, you must not change the content of your predictions.

Finally, please ensure you restructure the manuscript in the correct order for Articles in Communications Psychology: Abstract, Introduction, Methods, Results, Discussion. We recommend restructuring the sections before you address the reviewers' requests for changes in the presentation, as the order of the text will affect some of the changes you make. As you revise the formatting, please also ensure that your manuscript complies with all items on the linked checklist (below), including details about statistics reporting, separate Data and Code Availability statements and Acknowledgments.

Please use the following link to submit your revised manuscript, point-by-point response to the referees' comments (which should be in a separate document to any cover letter) and the completed checklist:

[Link redacted]

We hope to receive your revised paper within 8 weeks; please let us know if you aren't able to submit it within this time so that we can discuss how best to proceed. If we don't hear from you, and the revision process takes significantly longer, we may close your file. In this event, we will still be happy to reconsider your paper at a later date, provided it still presents a significant contribution to the

literature at that stage.

Please do not hesitate to contact me if you have any questions or would like to discuss these revisions further. We look forward to seeing the revised manuscript and thank you for the opportunity to review your work.

Best regards,

Antonia Eisenkoeck

Antonia Eisenkoeck
Senior Editor
Communications Psychology

EDITORIAL POLICIES AND FORMATTING

Editorial Policy: Policy requirements (Download the link to your computer as a PDF.)

Furthermore, please align your manuscript with our format requirements, which are summarized on the following checklist:

Communications Psychology formatting checklist

and also in our style and formatting guide Communications Psychology formatting guide .

* **CODE AVAILABILITY:** All Communications Psychology manuscripts must include a section titled "Code Availability" at the end of the methods section. In the event of publication, we require that the custom analysis code supporting your conclusions is made available in a publicly accessible repository; at publication, we ask you to choose a repository that provides a DOI for the code; the link to the repository and the DOI will need to be included in the Code Availability statement. Publication as Supplementary Information will not suffice. We ask you to prepare code at this stage, to avoid delays later on in the process.

* **DATA AVAILABILITY:**

All Communications Psychology manuscripts must include a section titled "Data Availability" at the end

of the Methods section or main text (if no Methods). More information on this policy, is available at <http://www.nature.com/authors/policies/data/data-availability-statements-data-citations.pdf>.

At a minimum the Data availability statement must explain how the data can be obtained and whether there are any restrictions on data sharing. Communications Psychology strongly endorses open sharing of data. If you do make your data openly available, please include in the statement:

We recommend submitting the data to discipline-specific, community-recognized repositories, where possible and a list of recommended repositories is provided at <http://www.nature.com/sdata/policies/repositories>.

If a community resource is unavailable, data can be submitted to generalist repositories such as figshare or Dryad Digital Repository. Please provide a unique identifier for the data (for example a DOI or a permanent URL) in the data availability statement, if possible. If the repository does not provide identifiers, we encourage authors to supply the search terms that will return the data. For data that have been obtained from publicly available sources, please provide a URL and the specific data product name in the data availability statement. Data with a DOI should be further cited in the methods reference section.

REVIEWERS' EXPERTISE:

Reviewer #1: cooperation & punishment, social network modelling

Reviewer #2: cooperation & punishment

Reviewer #3: cooperation & punishment in Colombian sample

REVIEWERS' COMMENTS:

Reviewer #1 (Remarks to the Author):

This paper reports on the results of economic experiments with two rural communities in Columbia, where the authors investigated whether direct and indirect reciprocity determines individuals' giving decisions to others in their community. The authors find evidence for both direct and indirect, and what they call generalized reciprocity: personal history with recipient matters, as does donors' perception of the recipient and the recipient's general standing.

The study seems well done; there is a pretty rich data set here. The questions are of long standing interest, and while the results are not particularly surprising, showing them robustly in two non-WEIRD populations is of considerably value. I think the study is worth publishing.

The authors should do some revisions, however, to improve the clarity of the manuscript. These are mainly cosmetic/expositional issues, but I think addressing them will make the paper more readable, so I'd encourage the authors to think about them:

Introduction: while nothing the authors say is outright wrong (though some I disagree with), I felt

that the presentation of the theoretical literature is a bit jumbled. For example, the authors assert that direct reciprocity “appears insufficient” to maintain cooperation, but actually in small communities like the ones they study direct reciprocity is entirely sufficient. This they assert on the way to arguing we need indirect reciprocity for human cooperation, while also dismissing institutional punishment as a mechanism (b/c of the free rider problem), but I think no one would argue with a straight face that these mechanisms are actually absent in human populations or an insignificant contributor to human cooperation. So, while again, this is maybe a matter of preference, I don’t understand why this argument is necessary: surely a more direct way to say that indirect reciprocity is a fact of human life, and therefore it’s important to understand how it works in different populations. Along those lines, some of the “drive by” references in the very beginning of the intro are a bit off: e.g., ref [5] for “inclusive fitness can maintain cooperation” goes to a recent paper that advocates a reinterpretation of IF (wrongly, IMO) instead of, say, Hamilton 1964 or Hamilton 1970 (the former is cited but the latter not). And surely there are papers before 2020 that dealt with kin discrimination/recognition. Perhaps more substantively, it is well known that kin selection doesn’t require kin recognition (you don’t say it does), and it is also well known (adding to the above point) that a little kin selection and a little reciprocity together goes a long way for cooperation.

My final comment for the introduction is that the conceptual framing is a bit disjoint from the predictions that follow, and the predictions themselves are then not mentioned ever again, either in the results section or the discussion. So, I’d suggest either not doing a numbered set of predictions, or actually organizing the results and perhaps also the discussion around them.

In terms of analysis and results: I think the authors underestimate how complex their analyses will be for the readers to take in. I felt I needed more structure, and hand-holding, as it were, when being walked through the analyses. A flip side of it is that the statistical sophistication (in some ways one of the main contributions) gets kinda buried. I think I understand the stats mostly, and don’t think anything is wrong with them, but I don’t think the authors make it easy on the reader. I would recommend a structured walk through of the Bayesian analytic strategies (a précis of what’s in the methods section) before going into the results, talking about what models were fit and what the interpretations are of the various fixed and random effects *before* launching into the results.

(A minor point here: you talk about generalized reciprocity but in a way that might cause some confusion, since the same phrase was used by Taborsky and colleagues to refer to a kind of reciprocity where the giver gives if it has been given before, not necessarily by the receiver, without regard to the receiver, their past history or whatever — a pay it forward kind of reciprocity. That threw me for a while, since that’s potentially also in your analysis. In any case you should make it a bit more explicit that’s not what you are talking about, and maybe consider using a different phrase, since you really think about indirect reciprocity based community standing, rather than individual standing.)

To reiterate, I think this is a neat paper, and I support its publication. I do think the authors can do a better job presenting the analytical strategy and results (and maybe the conceptual background, but that’s perhaps more subjective) and doing so would make the paper more accessible and impactful. These would amount to rewriting of some passages and adding or changing some structure, but should be easily doable.

Reviewer #2 (Remarks to the Author):

This paper examines key predictions from models of direct and indirect reciprocity using economic games within social networks of two rural Colombian communities. The findings overall support

theoretical predictions, showing that (1) community members condition their giving, exploitation, and punishment behavior on others' reputation; (2) reputations for generosity or selfishness track behavior in the economic games (although, to my understanding, reputations are not based on knowledge of this behavior); and (3) targets with generalized reputations for being generous are more likely to receive resources from community members (although they are not less likely to be punished), whereas targets with generalized reputations for being selfish are more likely to be exploited by community members.

The paper will undoubtedly be of interest to researchers across several fields studying cooperation, reputation, and punishment. The findings provide insights on the mechanisms promoting cooperation in social networks; the role of reputation in guiding cooperation, exploitation, and punishment decisions; and the social consequences of generalized reputations. The paper has many merits in terms of its methodology, including the use of incentivized decision-making tasks, in two communities in a non-WEIRD setting, and a rigorous social network approach to studying reputations. Methods are described in sufficient detail (though see some recommendations below) and the conclusions are warranted based on the findings.

Below, I provide some minor suggestions and recommendations to the authors

1. Already in the abstract and throughout the paper, the authors seem to use the terms 'standing' and 'reputation' interchangeably. Do they consider them interchangeable and, if so, why do they use both? If they are not interchangeable, could the authors provide definitions and briefly explain how standing differs from reputation?

2. The authors measure tendencies to make transfers toward those with a good reputation (i.e., positive reciprocity) as well as tendencies to exploit those with a bad reputation (i.e., negative reciprocity). Some previous work testing predictions from a 'strong reciprocity' perspective has found that individual tendencies for positive reciprocity are unrelated to tendencies for negative reciprocity (Weber, Weisel, & Gächter, 2018). Do the authors' data speak to this question of correlation between positive and negative reciprocity (see also point 4 below)?

3. A couple of points could be clarified in the description of the experimental tasks in the main text. After reading the methods, these points are very clear, but while reading the section on recipient identity-conditioned decisions and decider-anonymous reputations, I was confused about them. Probably this is just a matter of phrasing, but could the authors more explicitly mention there that (a) 'individuals know the identity of others during game play,' but this only applies to decision-makers and not recipients (if I understand correctly, receivers don't learn who gave them, punished them, etc.); and (b) reputations are not based on game play but they are based on participants' preexisting perceptions of targets?

4. Could the authors explain why they use the term 'spiteful punishment' in their results section? Is spiteful punishment taken to mean a cost infliction on another person at a personal cost but no personal benefit? Do the authors use the term just to refer to second-party punishment? Either way, it is not entirely clear here if punishment is motivated by spite (or by e.g., wanting to correct the behavior of a knowingly selfish other) and if it can be construed as second-party punishment (given the punisher has not necessarily been victimized).

Further, the authors state that 'punishment behavior' is associated with being perceived as selfish. This is a point of debate in the literature and I would suggest discussing in more detail how the authors' finding fits with prior work (some suggesting punishers are perceived more positively: e.g., Barclay, 2006; Jordan et al., 2016; and some suggesting punishers are perceived more negatively;

e.g., Eriksson et al., 2016; see also Raihani & Bshary, 2015).

Relatedly, the unexpected finding that individuals with generous reputations were either not less likely to be punished (in the lowland community) or were even more likely to be punished (in the coastal community) could be briefly discussed in the context of previous findings of antisocial punishment (e.g., Herrmann et al., 2008), e.g., before or after discussing the behavioral profile of punishers. Finally, I found this more detailed discussion of the profile of punishers in the lowland and coastal communities fascinating, also because there seems to be supportive evidence for a co-occurrence of prosociality and punishment in one community (lowland), but not the other (coastal) (see also comment 2 earlier).

References

- Barclay, P. (2006). Reputational benefits for altruistic punishment. *Evolution and Human Behavior*, 27(5), 325–344.
- Eriksson, K., Andersson, P. A., & Strimling, P. (2016). Moderators of the disapproval of peer punishment. *Group Processes & Intergroup Relations*, 19(2), 152–168.
- Herrmann, B., Thoni, C., & Gächter, S. (2008). Antisocial punishment across societies. *Science*, 319(5868), 1362–1367.
- Jordan, J. J., Hoffman, M., Bloom, P., & Rand, D. G. (2016). Third-party punishment as a costly signal of trustworthiness. *Nature*, 530(7591), 473–476.
- Raihani, N. J., & Bshary, R. (2015). The reputation of punishers. *Trends in Ecology & Evolution*, 30(2), 98–103.
- Weber, T. O., Weisel, O., & Gächter, S. (2018). Dispositional free riders do not free ride on punishment. *Nature Communications*, 9(1), 2390.

Reviewer #3 (Remarks to the Author):

>>>Reputation, exploitation, and the maintenance of cooperation: Evidence of direct and indirect reciprocity in network-structured economic games - Reviewer report<<<<

>>>Summary<<<

This manuscript reports the outcomes of three network-structured economic games, and how game decisions correlate with good and bad social standing, for two rural Colombian communities (340 individuals). Results using dyadic-level information and generalized reputation provide evidence of indirect reciprocity.

>>>Overview<<<

This study is successful in bridging social standing within two communities with decisions in three economic games of different nature: giving to others, taking from others—what the authors called “exploitation”—and punishing others. Fig. 2 makes very compelling the relationship between reputation (i.e., generous or selfish) and patterns of giving, taking, and punishment. Below, I provide a list of comments that may help the paper in terms of consideration of more specific

punishment mechanisms, methodological consistency, and clarity.

>>>Main comments<<<

>1. Anti-social punishment and the fear of counter-punishment might be playing a role that is not captured in the departing model.

Lines 130-139 describe a model where the interplay of reputation and network structure can sustain and maintain cooperation. An implicit assumption in this model is the absence of anti-social punishment.

This behavior should be mentioned in the paper because the reported results in lines 388-394, where individuals with generous reputations are not less likely to be punished in the lowlands, and more likely to be punished in the coastal community, may be interpreted in accordance with anti-social punishment behavior.

One possibility is to bring anti-social punishment in the discussion, as part of the aspects that are left out by Bhui et al.'s model on the expected role of punishment. It may also be connected to the proposed explanation for coastal punishers being exploitative and not generous.

The following references may also result helpful:

- Nikiforakis, N. (2008). Punishment and counter-punishment in public good games: Can we really govern ourselves?. *Journal of Public Economics*, 92(1-2), 91-112.

- Rand, D. G., Armao IV, J. J., Nakamaru, M., & Ohtsuki, H. (2010). Anti-social punishment can prevent the co-evolution of punishment and cooperation. *Journal of theoretical biology*, 265(4), 624-632.

- Balafoutas, L., Nikiforakis, N., & Rockenbach, B. (2014). Direct and indirect punishment among strangers in the field. *Proceedings of the National Academy of Sciences*, 111(45), 15924-15927.

>2. The formulation of P2 (lines 165-177) is confusing as it does not coincide with the game's informational structure.

Prediction P2a reads "Individual j will perceive individual k to be generous if k gave to j, and/or avoided exploiting/punishing j," suggesting that individual j observed individual k's action. This is confusing because it hints to the reader that participants had dyad-level information on who gives/exploits/punishes whom, which is not the case, as is later clarified in lines 202-205.

I suggest the authors to rewrite P2 such that it better reflects the informational structure of the RICH protocol.

Whereas the following lines are perhaps too simplistic, they helped me in grasping how P2 was tested:

P1a: "I give more to those I rate as more generous."

P2a: "I give more and then I am rated as more generous."

This comment applies similarly to P2b.

>3. Whereas the motivations for giving and exploiting behaviors are clear, it is not clear why participants should punish each other.

My main concern is an "experimenter demand effect" (see Zizzo, 2010; de Quidt et al., 2018): participants are allowed to deduce points from others at a cost to themselves, but there is no clear reason why they should do it. In most experimental paradigms involving punishment, some behavior from others is revealed. In absence of such information, as in this case, they may be punishing just because they are allowed to.

Which are my concerns? In absence of a clear reason to punish, participants may create their own informational cues. The benevolent interpretation is that they rely on reputation to make their punishment decisions. However, an alternative interpretation is that they rely on their own behavior in

the previous two games, and their expectations of others' behavior in these games. This is problematic because we know very little on why they are punishing, but also because the interdependence in the choices across the three games increases. To my understanding (but I may be missing something from the analytical strategy), this is not captured in the statistical analysis. My suggestion is to make a more thorough discussion regarding why participants should punish.

>4. The outcomes from the exploitation game are quite peculiar and deserve some additional discussion.

A rapid glance at Figs. 1 and 2, and Table 1, reveal that exploitation—or taking from others—was widespread. To the best of my knowledge this is quite uncommon (see List, 2007), so I wonder if this might be related to the population, to the experimental design, or to the protocol. I suggest the authors to expand their discussion regarding this point.

>5. Make explicit that the giving and exploiting networks cannot be directly compared.

By construction, participants took more choices with smaller stakes in the RICH exploiting than in the RICH giving game. I acknowledge that the authors never made a direct comparison, probably aware of the existing issues by doing so, but I consider important to clarify that part of the difference in the density of the networks and dyadic interactions across games appear by design.

>>>Minor comments<<<

>6. Lines 140-144: I agree that few studies exist that study networks of cooperation, exploitation, and punishment for non-WEIRD populations. Nonetheless, here are a couple of studies that you may find relevant:

- Attanasio, O., Barr, A., Cardenas, J. C., Genicot, G., & Meghir, C. (2012). Risk pooling, risk preferences, and social networks. *American Economic Journal: Applied Economics*, 4(2), 134-167.
- Chandrasekhar, A. G., Kinnan, C., & Larreguy, H. (2018). Social networks as contract enforcement: Evidence from a lab experiment in the field. *American Economic Journal: Applied Economics*, 10(4), 43-78.

>7. Lines 627-632: Perhaps you can add some references that sustain this statement.

>8. In the data collection section, please clarify whether all games were paid to all participants.

>9. The authors are kindly requested to share the following information:

A. Did the authors conceive a payment delivery method that minimizes the risk of violating the privacy of the amount earned by each participant? Please describe it.

B. Were the study co-authors of Colombian origin affiliated to any research institution in Colombia?

>>>References<<<

De Quidt, J., Haushofer, J., & Roth, C. (2018). Measuring and bounding experimenter demand. *American Economic Review*, 108(11), 3266-3302.

List, J. A. (2007). On the interpretation of giving in dictator games. *Journal of Political Economy*, 115(3), 482-493.

Zizzo, D. J. (2010). Experimenter demand effects in economic experiments. *Experimental Economics*, 13, 75-98.

Reputation, exploitation, and the maintenance of cooperation: Evidence of direct and indirect reciprocity in network-structured economic games: reply to reviewers

Daniel Redhead, Matthew Gervais, Kotrina Kajokaite, Jeremy Koster, Arlenys Hurtado Manyoma, Danier Hurtado Manyoma, Richard McElreath and Cody T. Ross

Dear Dr. Eisenkoeck,

We hope that this letter finds you well. We have now revised our manuscript and are resubmitting it to *Communications Psychology*. We would like to thank the reviewers and yourself for providing such informative and constructive feedback. We have tried to thoroughly address all comments made by the reviewers. These suggestions have certainly strengthened the manuscript a great deal.

The following are the most important changes that we have made to the manuscript:

- We have now restructured the paper to fit the formatting requirements of the journal. We believe that this reformatting has addressed many of the reviewer comments about clarity. Allowing us to introduce the methods first, so that readers know how to interpret the results presented in the manuscript.
- We have now added a 'limitations' section to the discussion.
- We have provided further justification of our predictions, and have made our predictions tie more closely to the introduction—ensuring that none of the predictions have changed.
- We have added some new data from two new sites, collected over the past year. The new data increases the power/generalizability of the study. However, none of the substantive results have changed. So we consider this a minor addition.

Alongside these major changes, we have made stylistic and formatting amendments, as requested by reviewers and yourself. Below is a point-by-point response.

Thanks again for your time and consideration,
Daniel Redhead and Cody T. Ross

REVIEWER # 1

COMMENT # 1.1

This paper reports on the results of economic experiments with two rural communities in Columbia, where the authors investigated whether direct and indirect reciprocity determines individuals' giving decisions to others in their community. The authors find evidence for both direct and indirect, and what they call generalized reciprocity: personal history with recipient matters, as does donors' perception of the recipient and the recipient's general standing.

Reply:

Thanks, this is a great overview of our main findings.

COMMENT # 1.2

The study seems well done; there is a pretty rich data set here. The questions are of long standing interest, and while the results are not particularly surprising, showing them robustly in two non-WEIRD populations is of considerably value. I think the study is worth publishing.

Reply:

Thank you very much for the positive feedback.

COMMENT # 1.3

The authors should do some revisions, however, to improve the clarity of the manuscript. These are mainly cosmetic/expositional issues, but I think addressing them will make the paper more readable, so I'd encourage the authors to think about them:

Reply:

Thank you for the feedback. We have now addressed all of the points that you raise. For clarity, we respond point-wise in the following text blocks.

COMMENT # 1.4

Introduction: while nothing the authors say is outright wrong (though some I disagree with), I felt that the presentation of the theoretical literature is a bit jumbled. For example, the authors assert that direct reciprocity "appears insufficient" to maintain cooperation, but actually in small communities like the ones they study direct reciprocity is entirely sufficient. This they assert on the way to arguing we need indirect reciprocity for human cooperation, while also

dismissing institutional punishment as a mechanism(b/c of the free rider problem), but I think no one would argue with a straight face that these mechanisms are actually absent in human populations or an insignificant contributor to human cooperation. So, while again, this is maybe a matter of preference, I don't understand why this argument is necessary: surely a more direct way to say that indirect reciprocity is a fact of human life, and therefore it's important to understand how it works in different populations.

Reply:

We have now slightly revised relevant sections of the introduction to tone down our language surrounding the importance of specific mechanisms for sustaining cooperation. For instance, we now emphasise that direct reciprocity appears insufficient to explain the *breadth of human cooperation, not that it isn't important in explaining cooperation*. In other words, there are context in which direct reciprocity is not sufficient to explain cooperation, even though there are contexts where it may be sufficient. We never meant to be dismissive of direct reciprocity, or kin selection. Both mechanism have robust empirical support. We simply add empirical support of indirect reciprocity.

Regarding the specific point that direct reciprocity is entirely sufficient in small-scale communities. We think that this point makes some sense. Certainly, these communities are far from being large, industrial cities with much totally anonymous interaction. However, they are large enough that reputation seems to matter. People give to and take from people outside their immediate social networks (i.e., the people they know well enough to judge on the basis of direct personal relationships), and this behavior is explained by aggregate level reputation ratings. Therefore, we have not included any suggestion that we believe that direct reciprocity alone can stabilize cooperation in these communities. Our results suggest scope for both direct and indirect reciprocity.

COMMENT # 1.5

Along those lines, some of the "drive by" references in the very beginning of the intro are a bit off: e.g., ref [5] for "inclusive fitness can maintain cooperation" goes to a recent paper that advocates a reinterpretation of IF (wrongly, IMO) instead of, say, Hamilton 1964 or Hamilton 1970 (the former is cited but the latter not). And surely there are papers before 2020 that dealt with kin discrimination/recognition. Perhaps more substantively, it is well known that kin selection doesn't require kin recognition (you don't say it does), and it is also well known (adding to the above point) that a little kin selection and a little reciprocity together goes a long way for cooperation.

Reply:

We believe that there is merit in Fromhage & Jennions' (2019) approach, and therefore retain the citation ([5]). Alongside this, we now include a citation to Hamilton (1970), and have re-worded our sentence on kin selection to make it clear that we are not stating that kin selection requires kin detection/discrimination. We have also added the following sentence to make clear that kin selection and direct reciprocity are still important mechanisms for sustaining cooperation:

" ..although inclusive fitness and direct reciprocity are important factors in sustaining cooperation, it seems that other mechanisms are still needed to explain the breadth of cooperation observed in many human groups"

COMMENT # 1.6

My final comment for the introduction is that the conceptual framing is a bit disjoint from the predictions that follow, and the predictions themselves are then not mentioned ever again, either in the results section or the discussion. So, I'd suggest either not doing a numbered set of predictions, or actually organizing the results and perhaps also the discussion around them.

Reply:

We have now moved our predictions into the section entitled: "An empirical analysis of direct and indirect reciprocity", and reworded some of the language in our predictions to ensure that the section flows more naturally from the introduction. We now explicitly refer to our predictions throughout the results section—noting (and hyper-linking to) the predictions that each of our results directly relates to.

COMMENT # 1.7

*In terms of analysis and results: I think the authors underestimate how complex their analyses will be for the readers to take in. I felt I needed more structure, and hand-holding, as it were, when being walked through the analyses. A flip side of it is that the statistical sophistication (in some ways one of the main contributions) gets kinda buried. I think I understand the stats mostly, and don't think anything is wrong with them, but I don't think the authors make it easy on the reader. I would recommend a structured walk through of the Bayesian analytic strategies (a précis of what's in the methods section) before going into the results, talking about what models were fit and what the interpretations are of the various fixed and random effects *before* launching into the results.*

Reply:

This is a very good point. We initially formatted this paper for another journal by the same publisher, which required methods to appear at the end. We have now restructured of the manuscript so that the methods section comes before the results.

COMMENT # 1.8

(A minor point here: you talk about generalized reciprocity but in a way that might cause some confusion, since the same phrase was used by Taborsky and colleagues to refer to a kind of reciprocity where the giver gives if it has been given before, not necessarily by the receiver, without regard to the receiver, their past history or whatever—a pay it forward kind of reciprocity. That threw me for a while, since that's potentially also in your analysis. In any case you should make it a bit more explicit that's not what you are talking about, and maybe consider using a different phrase, since you really think about indirect reciprocity based community standing, rather than individual standing.)

Reply:

We have now added a sentence in footnote 1 that clarifies our use of generalized reciprocity when we first use the term:

“We use the term generalized reciprocity in a way that remains in line with previous research that applies the social relations model, and should not be confused with previous uses of the term ‘generalized reciprocity’ in behavioural ecology that refers to situations where a giver allocates resources if it has been given before, or uses in economic anthropology that refer to situations where givers do not expect receivers to pay an equal amount of resources back in a predetermined time period”.

COMMENT # 1.9

To reiterate, I think this is a neat paper, and I support its publication. I do think the authors can do a better job presenting the analytical strategy and results (and maybe the conceptual background, but that's perhaps more subjective) and doing so would make the paper more accessible and impactful. These would amount to rewriting of some passages and adding or changing some structure, but should be easily doable.

Reply:

Thanks again for the positive feedback and the careful reading. We think that our edits should address all of the points raised here!

REVIEWER # 2

COMMENT # 2.1

This paper examines key predictions from models of direct and indirect reciprocity using economic games within social networks of two rural Colombian communities. The findings overall support theoretical predictions, showing that (1) community members condition their giving, exploitation, and punishment behavior on others' reputation; (2) reputations for generosity or selfishness track behavior in the economic games (although, to my understanding, reputations are not based on knowledge of this behavior); and (3) targets with generalized reputations for being generous are more likely to receive resources from community members (although they are not less likely to be punished), whereas targets with generalized reputations for being selfish are more likely to be exploited by community members.

The paper will undoubtedly be of interest to researchers across several fields studying cooperation, reputation, and punishment. The findings provide insights on the mechanisms promoting cooperation in social networks; the role of reputation in guiding cooperation, exploitation, and punishment decisions; and the social consequences of generalized reputations. The paper has many merits in terms of its methodology, including the use of incentivized decision-making tasks, in two communities in a non-WEIRD setting, and a rigorous social network approach to studying reputations. Methods are described in sufficient detail (though see some recommendations below) and the conclusions are warranted based on the findings.

Reply:

Thank you for this feedback.

COMMENT # 2.2

Below, I provide some minor suggestions and recommendations to the authors

Reply:

We address all of the points that you raise below.

COMMENT # 2.3

Already in the abstract and throughout the paper, the authors seem to use the terms 'standing' and 'reputation' interchangeably. Do they consider them interchangeable and, if so, why do they use both? If they are not interchangeable, could the authors provide definitions and briefly explain how standing differs from reputation?

Reply:

We made an active decision to use both ‘standing’ and ‘reputation’ throughout the manuscript. They are fairly similar terms: one referring to direct first-person perception of social standing, and one referring to perception of standing as influenced by third-party information/gossip. We do not use the terms interchangeably in most situations, unless the sentence is broad enough that a term like “standing/reputation” make sense. To make the distinction clearer in the manuscript, we have now added definitions to the following sentence:

“ In models of indirect reciprocity, individuals engage in costly cooperation or punishment in order to uphold their own standing (i.e., their dyadic standing based on first-person knowledge of past behavior) or reputation (i.e., their social standing based on aggregate third-party accounts and gossip)”

COMMENT # 2.4

The authors measure tendencies to make transfers toward those with a good reputation (i.e., positive reciprocity) as well as tendencies to exploit those with a bad reputation (i.e., negative reciprocity). Some previous work testing predictions from a ‘strong reciprocity’ perspective has found that individual tendencies for positive reciprocity are unrelated to tendencies for negative reciprocity (Weber, Weisel, & Gächter, 2018). Do the authors’ data speak to this question of correlation between positive and negative reciprocity (see also point 4 below)?

Reply:

We have pondered this. Our data might be useful to test such questions, but we think that we would need a different modeling approach than what we use here. The reviewer’s approach would require some kind of complex interaction model, that we do not yet know how to implement in the context of the SRM.

COMMENT # 2.5

A couple of points could be clarified in the description of the experimental tasks in the main text. After reading the methods, these points are very clear, but while reading the section on recipient identity-conditioned decisions and decider-anonymous reputations, I was confused about them. Probably this is just a matter of phrasing, but could the authors more explicitly mention there that (a) ‘individuals know the identity of others during game play,’ but this only applies to decision-makers and not recipients (if I understand correctly, receivers don’t learn who gave them, punished them, etc.);

Reply:

We have now moved the methods section to appear above the results section, which ensures greater clarity for readers when interpreting results. The following sentences now appear in the outline of the experimental games, and explicitly state that individuals know the identities of others during game play, but decisions remain “decider-confidential”:

“The games are recipient identity-conditioned—i.e., individuals know the identity of others during game play—but remain decider-confidential. In other words, individual j does not know—when rating individual k as selfish or generous—who individual k gave to, exploited, or punished.”

COMMENT # 2.6

and (b) reputations are not based on game play but they are based on participants’ preexisting perceptions of targets?

Reply:

We have added a sentence to outline that reputation was not based on game-play:

“Therefore, our measures of reputation are not based on game-play, but capture individuals’ pre-existing perceptions of other group members.”

COMMENT # 2.7

Could the authors explain why they use the term ‘spiteful punishment’ in their results section? Is spiteful punishment taken to mean a cost infliction on another person at a personal cost but no personal benefit? Do the authors use the term just to refer to second-party punishment? Either way, it is not entirely clear here if punishment is motivated by spite (or by e.g., wanting to correct the behavior of a knowingly selfish other) and if it can be construed as second-party punishment (given the punisher has not necessarily been victimized).

Reply:

For clarity, we have cut the words “spite” or “spiteful” from all sentences where they appeared. The confusion here seems to stem from psychologists and biologists using the word in different ways. Classic work like West and Gardner (2010. Altruism, Spite, and Greenbeards. *Science*) define “spite” as any behavior that reduces the fitness of a focal actor but decreases the fitness of another individual more. This is different than the definition of “spite” as an emotional state. For our purposes,

just saying “punishment” is sufficient. The addition of the word “spite” was just to emphasize that punishing was costly for the focal actor (i.e., that punishing others reduces a focal’s own payout).

COMMENT # 2.8

Further, the authors state that ‘punishment behavior’ is associated with being perceived as selfish. This is a point of debate in the literature and I would suggest discussing in more detail how the authors’ finding fits with prior work (some suggesting punishers are perceived more positively: e.g., Barclay, 2006; Jordan et al., 2016; and some suggesting punishers are perceived more negatively; e.g., Eriksson et al., 2016; see also Raihani & Bshary, 2015).

Reply:

These are great points, thanks! We have now added a brief outline of how our results speak to the mixed evidence found in the existing literature. We reviewed the suggested papers, and now cite them on lines 775:795.

COMMENT # 2.9

Relatedly, the unexpected finding that individuals with generous reputations were either not less likely to be punished (in the lowland community) or were even more likely to be punished (in the coastal community) could be briefly discussed in the context of previous findings of antisocial punishment (e.g., Hermann et al., 2008), e.g., before or after discussing the behavioral profile of punishers.

Reply:

We now mention that our results relating to punishment of generous individuals could be explained by antisocial punishment, and cite the suggested work on this topic on lines 860:870.

COMMENT # 2.10

Finally, I found this more detailed discussion of the profile of punishers in the lowland and coastal communities fascinating, also because there seems to be supportive evidence for a co-occurrence of prosociality and punishment in one community (lowland), but not the other (coastal) (see also comment 2 earlier)

Reply:

Thanks for the note. We have kept this info in the paper.

REVIEWER # 3

COMMENT # 3.1

This manuscript reports the outcomes of three network-structured economic games, and how game decisions correlate with good and bad social standing, for two rural Colombian communities (340 individuals). Results using dyadic-level information and generalized reputation provide evidence of indirect reciprocity.

Reply:

Thanks for this summary of our core findings.

COMMENT # 3.2

This study is successful in bridging social standing within two communities with decisions in three economic games of different nature: giving to others, taking from others—what the authors called “exploitation”—and punishing others. Fig. 2 makes very compelling the relationship between reputation (i.e., generous or selfish) and patterns of giving, taking, and punishment. Below, I provide a list of comments that may help the paper in terms of consideration of more specific punishment mechanisms, methodological consistency, and clarity.

Reply:

Thank you for your detailed feedback. We have now addressed all of the points that you raise below, and believe that your feedback has certainly improved the clarity and consistency of the manuscript.

COMMENT # 3.3

Anti-social punishment and the fear of counter-punishment might be playing a role that is not captured in the departing model.

Lines 130-139 describe a model where the interplay of reputation and network structure can sustain and maintain cooperation. An implicit assumption in this model is the absence of anti-social punishment. This behavior should be mentioned in the paper because the reported results in lines 388-394, where individuals with generous reputations are not less likely to be punished in the lowlands, and more likely to be punished in the coastal community, may be interpreted in accordance with anti-social punishment behavior. One possibility is to bring

anti-social punishment in the discussion, as part of the aspects that are left out by Bhui et al.'s model on the expected role of punishment. It may also be connected to the proposed explanation for coastal punishers being exploitative and not generous. The following references may also result helpful:

- Nikiforakis, N. (2008). *Punishment and counter-punishment in public good games: Can we really govern ourselves?*. *Journal of Public Economics*, 92(1-2), 91-112.
- Rand, D. G., Armao IV, J. J., Nakamaru, M., & Ohtsuki, H. (2010). *Anti-social punishment can prevent the co-evolution of punishment and cooperation*. *Journal of theoretical biology*, 265(4), 624-632.
- Balafoutas, L., Nikiforakis, N., & Rockenbach, B. (2014). *Direct and indirect punishment among strangers in the field*. *Proceedings of the National Academy of Sciences*, 111(45), 15924-15927.

Reply:

This is a great point, thanks! In line with both yours and Reviewer 2's feedback, we have added a small explanation of our findings in relation to antisocial punishment. We have also included the suggested citations on lines 860:870. We didn't directly bring up the fact that the Bhui et al. model left out these details, however, as such details are a bit more nuanced than our empirical analysis permits us comment on.

COMMENT # 3.4

The formulation of P2 (lines 165-177) is confusing as it does not coincide with the game's informational structure. Prediction P2a reads "Individual j will perceive individual k to be generous if k gave to j, and/or avoided exploiting/punishing j," suggesting that individual j observed individual k's action. This is confusing because it hints to the reader that participants had dyad-level information on who gives/exploits/punishes whom, which is not the case, as is later clarified in lines 202-205. I suggest the authors to rewrite P2 such that it better reflects the informational structure of the RICH protocol. Whereas the following lines are perhaps too simplistic, they helped me in grasping how P2 was tested:

P1a: "I give more to those I rate as more generous."

P2a: "I give more and then I am rated as more generous."

This comment applies similarly to P2b.

Reply:

This is a good point. In the first draft, we first stated the predictions generally, as they were derived from the general Bhui et al. model. Then, we mentioned our operationalization of the predictions. The reviewer is right that this just leads to confusion. We now introduce the operationalized predictions directly, stating that: "...perceived standing will be influenced by cooperative behavior, with individual j's perception

of individual k tracking how k treats j on an ongoing basis. Assuming that behavior in the economic games parallels or proxies for behavior in real-world contexts (i.e., that individual j is more likely to give to individual k in the allocation game if j gives to, or shares with, k in real-world contexts; see Pisor et al. [20] for evidence from Colombia), this leads to predictions that...". In other words, we make it clear that prediction 2 is only expected to work if behavior in the games proxies for behavior in real-world settings (which is the case in for the RICH games in Colombia).

COMMENT # 3.5

Whereas the motivations for giving and exploiting behaviors are clear, it is not clear why participants should punish each other. My main concern is an "experimenter demand effect" (see Zizzo, 2010; de Quidt et al., 2018): participants are allowed to deduce points from others at a cost to themselves, but there is no clear reason why they should do it. In most experimental paradigms involving punishment, some behavior from others is revealed. In absence of such information, as in this case, they may be punishing just because they are allowed to. Which are my concerns? In absence of a clear reason to punish, participants may create their own informational cues. The benevolent interpretation is that they rely on reputation to make their punishment decisions. However, an alternative interpretation is that they rely on their own behavior in the previous two games, and their expectations of others' behavior in these games. This is problematic because we know very little on why they are punishing, but also because the interdependence in the choices across the three games increases. To my understanding (but I may be missing something from the analytical strategy), this is not captured in the statistical analysis. My suggestion is to make a more thorough discussion regarding why participants should punish.

Reply:

This is a good point. Pay-off maximizers shouldn't punish at all (if they are economically rational). Nothing about the game structure should motivate punishment. This is not a bug, but a purposeful feature of the RICH games.

Participants, however, have a history of interactions with other participants. We believe that this history creates deep interpersonal sentiments between individuals that are brought into the game. Through analysis of punishment behavior as a function of focal, alter, and dyadic covariates, we can identify the potential drivers of spite (or—to avoid the word spite—negatively-valanced connections). In Gervais (2017) and Pisor et al. (2019), we analyze the predictors of punishment, finding that punishment is often used against wealthier alters as a form of leveling. However, other causes (e.g., infidelity) were mentioned in qualitative interviews.

Here, we are simply measuring the extent to which punishment proclivities covary with standing/reputation ratings. We are not necessarily interested in explaining *why* we see covariance between punishment and standing/reputation, rather we are just estimating what that covariance is. It is important to note, however, that there is no reason for individuals to “rely on their own behavior in the previous two games” or on “their expectations of others’ behavior in these games” since the RICH games were designed such that the game structure provides no incentive for using that information as a basis for punishment. No matter what the other people do in the game, each person will lose money by punishing. And each player makes their own decisions in private, so punishment is not expected to influence overall cooperation rates, as it might in a public goods game with punishment, for example.

COMMENT # 3.6

The outcomes from the exploitation game are quite peculiar and deserve some additional discussion. A rapid glance at Figs. 1 and 2, and Table 1, reveal that exploitation—or taking from others—was widespread. To the best of my knowledge this is quite uncommon (see List, 2007), so I wonder if this might be related to the population, to the experimental design, or to the protocol. I suggest the authors to expand their discussion regarding this point.

Reply:

We have added the citation you suggest. We think that the design of the RICH games creates a fairly large scope for taking. Each person has the option to take from each and every other person in the community, including people they barely know. Decisions are also made in private. Thus, the scope for taking from others is very large. We mention this when discussing the density of the taking networks. We think that it is pretty hard to compare taking behavior in the RICH games to taking behavior in other paradigms.

COMMENT # 3.7

Make explicit that the giving and exploiting networks cannot be directly compared. By construction, participants took more choices with smaller stakes in the RICH exploiting than in the RICH giving game. I acknowledge that the authors never made a direct comparison, probably aware of the existing issues by doing so, but I consider important to clarify that part of the difference in the density of the networks and dyadic interactions across games appear by design.

Reply:

This is mostly correct. It is very difficult to compare base-rates and raw data on giving and taking, as the game set-up (i.e., number of coins, and coin value) is different.

This being said, our inferential statistical model does allow us to compare giving and exploitation networks, as well as any other kind of network, even networks produced using other methods. The SRM structure is designed to control for base-rate differences, as well as differences in the distribution of node-level in- and out-degree, permitting comparisons in reciprocity measures conditional on accounting for these differences in network structure. We make it clear in Table 1 that key network properties differ between different network layers. In the methods, we make it clear that each layer was produced using a different methodology. Finally, in the modeling section, we show how the SRM provides a framework for the joint modeling of all five network layers, which accounts for variation in base-rate, node-level properties, and dyad level properties across layers. This should address the reviewer's concern here.

COMMENT # 3.8

6. Lines 140-144: I agree that few studies exist that study networks of cooperation, exploitation, and punishment for non-WEIRD populations. Nonetheless, here are a couple of studies that you may find relevant:

- Attanasio, O., Barr, A., Cardenas, J. C., Genicot, G., & Meghir, C. (2012). Risk pooling, risk preferences, and social networks. *American Economic Journal: Applied Economics*, 4(2), 134-167.

- Chandrasekhar, A. G., Kinnan, C., & Larreguy, H. (2018). Social networks as contract enforcement: Evidence from a lab experiment in the field. *American Economic Journal: Applied Economics*, 10(4), 43-78.

Reply:

Many thanks for the references! We have added them in the appropriate place (now line 155).

COMMENT # 3.9

Lines 627-632: Perhaps you can add some references that sustain this statement.

Reply:

We have now added a citation that supports this statement.

COMMENT # 3.10

In the data collection section, please clarify whether all games were paid to all participants.

Reply:

We now state: "After all interviews in a site were completed, every participant was paid the total lump sum they earned across all games in which they were a party;"

COMMENT # 3.11

The authors are kindly requested to share the following information:

A. Did the authors conceive a payment delivery method that minimizes the risk of violating the privacy of the amount earned by each participant? Please describe it.

Reply:

We now state: "...payments were delivered in private, in sealed envelopes."

COMMENT # 3.12

B. Were the study co-authors of Colombian origin affiliated to any research institution in Colombia?

Reply:

No, neither Colombian co-authors were affiliated with a Colombian research institution; their only affiliation is the Max Planck Institute for Evolutionary Anthropology.

Decision letter and referee reports: second round

2nd Nov 23

Dear Dr Redhead,

Thank you for your patience during the peer-review process and apologies for the delay in reaching a decision. We sought advice from an additional reviewer with expertise in Bayesian modelling in response to the addition of new data.

As such, your manuscript titled "Reputation, exploitation, and the maintenance of cooperation: Evidence of direct and indirect reciprocity in network-structured economic games" has now been seen by 4 reviewers, and I include their comments at the end of this message. The initial three reviewers are happy with your revisions, however, the additional reviewer has identified some shortcomings that we hope you will be able to address before we make a final decision. We remain interested in the possibility of publishing your study in Communications Psychology, but would like to consider your responses to these concerns and assess a revised manuscript before we make a final decision on publication.

We therefore invite you to revise and resubmit your manuscript, along with a point-by-point response to the reviewers. Please highlight all changes in the manuscript text file.

Specifically, Reviewer #4 asks for some textual clarifications/revisions and for the implementation of an analysis method that controls for (potential) collinearity.

Please note that your revised manuscript must comply with our formatting and reporting requirements, which are summarized on the following checklist: Communications Psychology formatting checklist and also in our style and formatting guide Communications Psychology formatting guide .

Please use the following link to submit your revised manuscript, point-by-point response to the referees' comments (which should be in a separate document to any cover letter) and the completed checklist:

[Link redacted]

Please do not hesitate to contact me if you have any questions or would like to discuss these revisions

further. We look forward to seeing the revised manuscript and thank you for the opportunity to review your work.

Best regards,

Antonia Eisenkoeck

Antonia Eisenkoeck
Senior Editor
Communications Psychology

EDITORIAL POLICIES AND FORMATTING

Editorial Policy: Policy requirements (Download the link to your computer as a PDF.)

* **CODE AVAILABILITY:** All Communications Psychology manuscripts must include a section titled "Code Availability" at the end of the methods section. In the event of publication, we require that the custom analysis code supporting your conclusions is made available in a publicly accessible repository; at publication, we ask you to choose a repository that provides a DOI for the code; the link to the repository and the DOI will need to be included in the Code Availability statement. Publication as Supplementary Information will not suffice. We ask you to prepare code at this stage, to avoid delays later on in the process.

* **DATA AVAILABILITY:**

All Communications Psychology manuscripts must include a section titled "Data Availability" at the end of the Methods section or main text (if no Methods). More information on this policy, is available at <http://www.nature.com/authors/policies/data/data-availability-statements-data-citations.pdf>.

At a minimum the Data availability statement must explain how the data can be obtained and whether there are any restrictions on data sharing. Communications Psychology strongly endorses open sharing of data. If you do make your data openly available, please include in the statement:

We recommend submitting the data to discipline-specific, community-recognized repositories, where possible and a list of recommended repositories is provided at <http://www.nature.com/sdata/policies/repositories>.

If a community resource is unavailable, data can be submitted to generalist repositories such as figshare or Dryad Digital Repository. Please provide a unique identifier for the data (for example a DOI or a permanent URL) in the data availability statement, if possible. If the repository does not provide identifiers, we encourage authors to supply the search terms that will return the data. For data that have been obtained from publicly available sources, please provide a URL and the specific data product name in the data availability statement. Data with a DOI should be further cited in the methods reference section.

REVIEWERS' COMMENTS:

Reviewer #1 (Remarks to the Author):

I think my comments (which were relatively minor anyway) have been mostly addressed. I think this is a neat study worth publishing.

Reviewer #2 (Remarks to the Author):

Thank you for inviting me to review the revised version of this manuscript. The authors have addressed all my comments on the previous version, and I think the manuscript has improved. In particular, moving the methods section helps a great deal with following the study setup and results. I thank the authors for their responsiveness to the other issues raised. I have no further suggestions.

Reviewer #3 (Remarks to the Author):

All my concerns have been clarified. I thank the authors for the time taken to address my comments, and for producing a very organized and easy to read rebuttal letter.

Reviewer #4 (Remarks to the Author):

When I first received the review invitation, I was really looking forward to reading this paper. I am a Bayesian modeler, and although I am a cognitive psychologist, I have a degree in cultural anthropology. I was looking forward to reading this paper and hoping for a tight articulation between question and method. But I have run into substantial conceptual difficulties that preclude my understanding.

As I understand it, there are three hypotheses in question. P1: People reward those they directly rate as generous and punish/exploit those they directly rate as selfish. P2: Ratings reflect gameplay in an online, dynamic manner. P3: People reward those that others rate are generous and punish/exploit those that others rate are selfish.

My main concern is collinearity between P1 and P3. Assuredly, if a person is truly generous, there will be large correlations across dyadic and generalized ratings. Collinearity seems to be ignored and the problem is especially difficult in a posterior estimation framework such as that used here. Perhaps a very simple example will suffice. Suppose I have a set of wooden nesting dolls and I construct a linear model of the doll's girth as a function of the doll's height and weight. Clearly, there is collinearity, but I can estimate separate significant effects of height and weight and conclude that both have an effect on girth even though one is essentially a replicate of the other. Even more alarming, the effects I estimate will depend entirely on the ratio of prior variances. My own sense is that this problem necessitates a reasonable model comparison approach rather than estimation so that the amount of unique explanatory power of each covariate may be assessed. I was quite dismayed to see no discussion of the issue.

Here are other spots where my understanding was compromised:

- I had trouble identifying how others view good and bad standing were operationalized in P3, but it seems to be marginal across all ratings (see line 540). I should not have to search so diligently for it and it should not be so late. Carrying this ambiguity so long affected my reading of the paper.

- I think the form in (4) is awkward. I had to first translate to a standard quadratic form with the correlation written as $L'L$ and think long-and-hard about why it is so and remind myself the goal is to estimate the correlation $L'L$. It is not obvious at all. It becomes a bit more obvious for dyadic terms, but please, a little more support for the reader would be helpful. Since the goal is to estimate correlations, I would bury the Cholesky version in a footnote or refer to the stan manual, and keep the focus on the correlations. Also, the $*$ symbol is undefined.

- With regard to P2, I had trouble understanding how gameplay could be anonymous and yet affect dynamic ratings. Doesn't one need a stochastic process model here to understand updating?

- I am so confused about ρ . I thought it was a matrix with dimensionality of J^2 -by- J^2 for each m with a particular structure in (14). Now, starting at line 575, it seems to be a scalar with a posterior mean and CI. I am so lost that I stopped here.

Take home: This paper has the potential to succeed as it uses cutting edge methods to address important questions. Nonetheless, I think there are huge gaps. Perhaps I am too far a field to understand this paper. My sense, however, is that many others will struggle. The logic and articulation between question and method is not transparent.

REVIEWERS' COMMENTS:

Reviewer #1 (Remarks to the Author):

I think my comments (which were relatively minor anyway) have been mostly addressed. I think this is a neat study worth publishing.

Thanks for taking the time to give our work a second read, we are glad to hear that you believe our work to be of publishable standard.

Reviewer #2 (Remarks to the Author):

Thank you for inviting me to review the revised version of this manuscript. The authors have addressed all my comments on the previous version, and I think the manuscript has improved. In particular, moving the methods section helps a great deal with following the study setup and results. I thank the authors for their responsiveness to the other issues raised. I have no further suggestions.

Thanks for the feedback. The paper does read much better after reorganizing.

Reviewer #3 (Remarks to the Author):

All my concerns have been clarified. I thank the authors for the time taken to address my comments, and for producing a very organized and easy to read rebuttal letter.

Thanks again.

Reviewer #4 (Remarks to the Author):

When I first received the review invitation, I was really looking forward to reading this paper. I am a Bayesian modeler, and although I am a cognitive psychologist, I have a degree in cultural anthropology. I was looking forward to reading this paper and hoping for a tight articulation between question and method. But I have run into substantial conceptual difficulties that preclude my understanding.

We have tried our best to tightly link our research questions, game design, and statistical methodology. The statistical models we use are, in fact, custom built, bespoke models, that are designed specifically for studying the multi-layer structure of the economic game and standing-perception networks. We have re-read the manuscript in an attempt to ensure that our questions and methods are laid out in a coherent manner, and we believe that they are.

As I understand it, there are three hypotheses in question. P1: People reward those they directly rate as generous and punish/exploit those they directly rate as selfish. P2: Ratings reflect gameplay in an online, dynamic manner. P3: People reward those that others rate are generous and punish/exploit those that others rate are selfish.

It seems like the reviewer's understanding of P2 is incorrect. Ratings are **not** based on game-play, and **nothing was done online**. We conducted research in four rural Colombian communities and asked

people to privately identify those individuals in their communities they thought were the most selfish and the most generous. We actually went out to these communities, in person, and spent several months living in each, in order to collect the data used in this study. We presented photo-rosters containing pictures of all adult community members to each respondent, and each respondent indicated who was especially generous or selfish by placing green and purple tokens on the physical roster of pictures. If individual *j* placed a purple token on individual *k*, that would be recorded as a dyadic rating with individual *j* characterizing individual *k* as a selfish person. We describe this all, in detail, in the Methods section. See lines 300-332 and 360-370.

It seems like part of the reviewer's confusion is owed to the reviewer misinterpreting the word "dyadic" as "dynamic". The reviewer makes the same misinterpretation later when they says "I had trouble understanding how gameplay could be anonymous and yet affect dynamic ratings". The word dyadic means: "relating to the interaction between two individuals," in contrast to the word dynamic, which refers to: "changes over time in forces that control individuals or objects". We call the ratings *dyadic* because they represent directed network ties (i.e., sentiments) between *two* individuals. We did an electronic search of our paper, to check and see if we mistakenly wrote "dynamic" instead of "dyadic" anywhere. We did not.

My main concern is collinearity between P1 and P3. Assuredly, if a person is truly generous, there will be large correlations across dyadic and generalized ratings.

All ratings were **dyadic**, as we described in the Methods section. We did a search, and we never used the phrase "generalized ratings". We never collected "generalized ratings". As such, there can be no correlations/collinearity between dyadic and generalized ratings. Generalized rating do not exist. Theoretically, there is no reason to suppose that generosity must a stable, individual-level property. In fact, psychologists (see Gervais and Fessler 2017¹) typically view such characterizations as informing dyadic interpersonal sentiment: I can be generous towards some people (e.g., my close kin) and be selfish or stingy with others (e.g., my non-kin). This would lead kin to rate me as generous, and non-kin to rate me as selfish. Somebody else might be more broadly generous (perhaps they read a lot of Kant), and so they provide aid irrespective of kinship, leading to elevated generosity ratings by many others. Both patterns are logically possible, and the model that we used was developed specifically to disentangle such patterns. We literally partition variance into components explained by dyadic features, and components explained by general, individual-level features.

To be generous to the reviewer, the probability of individual *j*, for example, giving to individual *k*, may be predicted to vary as a function of both: 1) *j*'s perception of *k* as selfish, and 2) *k*'s overall tendency to be viewed as selfish. However, we use a well-tested model—the social relations model (SRM) Kenny and LaVoie (1984)²—to do this. We did not make this model up. It has been used with much success, in sociology, psychology, and applied statistics for close to 40 years, and it was developed precisely to separate these two kinds of effects. The sentence that the reviewer starts with "Assuredly" is simply wrong, both statistically and empirically. Empirically, in this dataset, it is quite common for some people to be widely viewed as generous. However, a few particular people in the community

¹ Gervais, M. M., & Fessler, D. M. (2017). On the deep structure of social affect: Attitudes, emotions, sentiments, and the case of "contempt". *Behavioral and Brain Sciences*, 40, e225.

Fessler, D. M., & Gervais, M. (2010). From whence the captains of our lives: Ultimate and phylogenetic perspectives on emotions in humans and other primates. *Mind the gap: Tracing the origins of human universals*, 261-280.

² Kenny, D. A., & La Voie, L. (1984). The social relations model. In *Advances in experimental social psychology* (Vol. 18, pp. 141-182). Academic Press.

might rate those same people as selfish. Why? Because those central, particularly generous people may have essentially shunned or ostracized a few particular people in the community. It is not uncommon to see, for example, people with substance or alcohol abuse problems rate the most generous people in a community as selfish, because those generous people help others, but not those who would use the money to buy alcohol. If we were to predict dyadic giving simply on the basis of generalized standing as generous, we would make bad predictions. By factoring in data on perceptions of selfishness, we make get more accurate predictions concerning dyadic giving, especially for those individuals whose perceptions conflict with the rest of the community.

Statistically speaking, the social relations model uses an incredibly clever pair of correlated random effects structures to separate out dyadic correlations (i.e., correlations in flows from j to k , and k to j) from “generalized” correlations (i.e., correlation between the node/individual-level in-degree and out-degree of individuals). We will not go into details here, but we refer the reviewer to classic papers (Kenny and LaVoie 1984), newer papers (Koster et al. 2020, Redhead et al. 2023, Ross et al. 2023³), as well as introductory Bayesian Statistics textbooks (e.g., see section 14.4 of Statistical Rethinking⁴, written by our most senior coauthor) for details. Again, the SRM has a near half-century history of being used specifically to disentangle generalized and dyadic correlations, our extension to the SRM further expands the power of the method, permitting estimation of the same effects both within and between network layers.

The multi-layer SRM structure that we use is not affected by collinearity issues of the kind the reviewer alludes to. Below, we will prove that more formally. Here, let us simply provide a set of example graphs:

Figure 1 :

Note that in figure (1a), every node has an in-degree of 1, and an out-degree of 1. Thus, generalized reciprocity is 0 (there is no variation in nodal degree, so there can't be covariation either). Likewise,

³ Koster, J., Leckie, G., & Aven, B. (2020). Statistical methods and software for the multilevel social relations model. *Field Methods*, 32(4), 339-345.

Redhead, D., McElreath, R., & Ross, C. T. (2023). Reliable network inference from unreliable data: A tutorial on latent network modeling using STRAND. *Psychological Methods*.

Ross, C. T., McElreath, R., & Redhead, D. (2023). Modelling animal network data in R using STRAND. *Journal of Animal Ecology*.

⁴McElreath, R. (2018). *Statistical rethinking: A Bayesian course with examples in R and Stan*. Chapman and Hall/CRC. <https://github.com/Booleans/statistical-rethinking/blob/master/Statistical%20Rethinking%202nd%20Edition.pdf>

dyadic reciprocity is null—no dyadic ties are reciprocated. Now look at figure (1b), every node has an in-degree of 1, and an out-degree of 1. Thus, generalized reciprocity is again 0. However, now, dyadic reciprocity is maximal—every single dyadic tie is reciprocated. Thus, the reviewer’s claim that these two things must be co-linear is wrong.

More generally, one can use our network analysis package:

https://github.com/ctross/STRAND/blob/main/R/simulate_srm_network.R

to simulate data from the social relations model, using arbitrary combinations of values for dyadic and generalized reciprocity. Then, one can use our analysis function:

<https://github.com/ctross/STRAND/blob/main/R/fit_srm_model.R>

to recover the parameters used to simulate the data. We provide such tests in the supplementary appendices of Redhead et al. 2023 and Ross et al. 2023, proving that the SRM as we have coded it permits accurate estimation of both generalized and dyadic correlations.

Collinearity seems to be ignored and the problem is especially difficult in a posterior estimation framework such as that used here. Perhaps a very simple example will suffice. Suppose I have a set of wooden nesting dolls and I construct a linear model of the doll's girth as a function of the doll's height and weight. Clearly, there is collinearity, but I can estimate separate significant effects of height and weight and conclude that both have an effect on girth even though one is essentially a replicate of the other. Even more alarming, the effects I estimate will depend entirely on the ratio of prior variances.

This is a great example of collinearity, and one that is very similar to the example that our senior coauthor (RM) uses in his textbook, *Statistical Rethinking*—he uses measurements of left and right legs length as a predictor of height. Even more collinear, but equally funny.

The reviewer gets a few things wrong here, though. First, in most people, both left-leg length and right-leg length contain information about height. Collinearity arises as a potential problem when both right and left leg length are used in the same model, and because both measures contain close to the same information, the marginal effect of left-leg length conditional on right-leg length is small (and vice versa), even though we know that leg length impacts height. The problem is that collinearity might lead us to conclude that there is no relationship between height and leg length, even though we know there is. The reviewer seems to get this wrong, suggesting somehow that it’s a flaw that both height and weight covary with girth in univariate models. It is not a flaw. The flaw comes from *including both in the same model, and then concluding that neither covaries with girth.*

Second, is the shocking claim that collinearity is “especially difficult” in Bayesian estimation frameworks using MCMC. In Bayesian frameworks, collinearity can be so high that you literally include the same predictor variable twice, and you can still get perfectly behaved models (See code example 5.29 in *Statistical Rethinking* to go through a worked example). However, you will only be able to identify the sum the coefficients. Either coefficient will be essentially “unidentified” (and the posterior distribution will have utterly enormous credible regions if you start with flat priors like we do here), but the overall model will be identified. In Bayesian analysis, *if you have proper priors then you will have a proper posterior* (with a Lebesgue measure 0 set of exceptions). As long as the posterior distribution is proper—which just means that it integrates to 1—then all of the parameters are identified. In maximum likelihood estimation procedures, however, serious problems can emerge when collinearity in predictors is high, standard errors can blow up to infinity, and models will not converge—rendering all parameter estimates from the model meaningless. We are not really sure what the statement “the effects I estimate will depend entirely on the ratio of prior variances” means. It certainly is incorrect: it is impossible for the statement to be true given Bayes’ rule. If parameter

estimates are conditioned on data, then it cannot possibly be true that they depend only on prior variances, they also depend on the data. Even if you have perfect collinearity in predictors, the sum of the coefficients is constrained by the data, and hence impacts the posterior in some way. We recommend the reviewer take a look at Section 5.3.1. “Multicollinear legs” of Statistical Rethinking⁵, or similar texts in other books.

To get more to the point, collinearity is a problem in models **with multiple predictor variables**, like this:

$$Y_i \sim \text{Normal}(\mu_i, \text{Sigma})$$

where:

$$\mu_i = a + b \cdot X_i + c \cdot Z_i$$

and where X and Z are tightly correlated.

Imagine that the correlation between X and Z goes to 1, such that $X=Z$. Then, then we can write:

$$\mu_i = a + b \cdot X_i + c \cdot X_i$$

which factors out as:

$$\mu_i = a + (b+c) \cdot X_i$$

and so only the sum of coefficients (b+c) can be identified.

If predictors are collinear, then the posterior variances on b and c would be enormous (as there are infinite combinations of two parameters that sum to the same value). If something like this was a problem for us, then we would see that the estimates in Figures 3 and 4 have huge standard errors. **We don't, so it's not an issue.**

Perhaps more striking, we don't use a regression with multiple predictors, so collinearity of predictors is not even possible in theory. As is readily apparent, by looking at the model description we provide on lines 390-540, there is no part of our model where we include **any** variable as a predictor of another variable! We have nothing of the form: $Y \sim X + Z$. So where does the reviewer think the collinearity in predictors arises? The reviewer certainly did not see us write out an equation of the type where such collinearity could arise. It looks like the reviewer did not read the model very closely. We do not use a model structure where such a phenomena could even possibly occur. (This, coincidentally, is why we do not discuss the issue in the paper).

If we were naive, and wrote our model like this:

$$\text{Give}_{j,k} \sim \text{Normal}(a + b \cdot \text{Selfish}_{j,k} + c \cdot \text{Generous}_{j,k} + \dots, \text{Sigma})$$

then collinearity would have been something to look out for. In fact, our awareness of such a potential issue, is why we did not use the single-layer SRM with peer-ratings layers as predictors of game-play ties, and why instead developed the multiplex generalization of the SRM.

A casual look by a competent reader, at either our math in Eqs. 1-17, or our results in Figure 3 and 4 would be enough to dismiss the question about collinearity.

My own sense is that this problem necessitates a reasonable model comparison approach rather than estimation so that the amount of unique explanatory power of each covariate may be assessed. I was quite dismayed to see no discussion of the issue.

⁵ <https://civil.colorado.edu/~balajir/CVEN6833/bayes-resources/RM-StatRethink-Bayes.pdf>

There is no reason to be dismayed. Given that collinearity is only possible when two predictors of an outcome are colinear, and that we don't use a regression with multiple predictors, it's not an issue that that we were concerned about.

The reviewer simply fails to understand how the social relations model works. We do not use any of our data as "covariates" to predict another. Instead, we model all five network layers as outcomes that are generated by (potentially) correlated random effects. Again, we can do nothing except refer the reviewer to the literature on the social relations model. We model the data appropriately.

Here are other spots where my understanding was compromised:

- I had trouble identifying how others view good and bad standing were operationalized in P3, but it seems to be marginal across all ratings (see line 540). I should not have to search so diligently for it and it should not be so late. Carrying this ambiguity so long affected my reading of the paper.

There was no need to carry the ambiguity so long. **We define our precise operationalization much earlier in the Methods section of the text, on lines 360–370:**

“...dyadic peer ratings for social standing/reputation were elicited by asking participant to place tokens on the photographs of community members who were especially generous (green tokens) or selfish (purple tokens). There was no minimum or maximum limit on the number of tokens that could be placed by each respondent, but most respondents used around 7 to 9 tokens per rating category.”

A careful reading of the above lines would show that we have independent assessments of generosity and selfishness, each at a dyadic level. A careful reading of our statistical model and results would show that we analyze these dyadic outcome measures as distinct network layers, while studying the structure of the correlated random effects that generate them (and all other network layers) at both a dyadic level and a generalized (individually aggregated) level. See Figures 3 and 4.

- I think the form in (4) is awkward. I had to first translate to a standard quadratic form with the correlation written as $L'L$ and think long-and-hard about why it is so and remind myself the goal is to estimate the correlation $L'L$. It is not obvious at all. It becomes a bit more obvious for dyadic terms, but please, a little more support for the reader would be helpful. Since the goal is to estimate correlations, I would bury the Cholesky version in a footnote or refer to the stan manual, and keep the focus on the correlations.

Although we appreciate that most quantitative social scientists initially learn how to write hierarchical models using simple notation, like:

$$\boldsymbol{\eta} \sim \text{Multivariate Normal}(\boldsymbol{\mu}, \boldsymbol{\Sigma})$$

and are taught to put rather problematic priors (like inverse Wishart) directly on the covariance matrix, $\boldsymbol{\Sigma}$, there has been ever increasing consensus—led by leaders in the field of Bayesian statistics, like Andrew Gelman—that such model definitions often lead to botched analyses, either because the prior

ends up dominating the posterior when it shouldn't (Gelman 2006⁶), or because numerical issues lead to poor model performance (Lewandowski et al. 2009, Stan User Guide 2023⁷).

A lot of really great work has been done by statisticians to both develop better priors for multilevel models (Lewandowski et al. 2009), and better, more numerically stable ways, of defining mathematically equivalent models (Stan User Guide 2023). We follow the latest suggestions for best practices when coding multi-level models, and write the equation above as:

$$\boldsymbol{\beta} = \boldsymbol{\mu} + \boldsymbol{\sigma} \circ (\mathbf{L}^* \boldsymbol{\eta})$$

where $\boldsymbol{\sigma}$ is a vector of standard deviations, $\mathbf{L}^* \mathbf{L}' = \boldsymbol{\rho}$ (where $\boldsymbol{\rho}$ is a correlation matrix), and where the elements of $\boldsymbol{\eta}$ have unit-normal—i.e., Normal(0,1)—priors. We have gone back to the text here, however, and made a few small rewrites for clarity.

Importantly, Eq.1 and Eq.2 above are mathematically equivalent (see Stan User Guide 2023 for proof), but they are not computationally equivalent. We believe in setting a good example for future scholars, by explicitly showing how to write a model in a form that is computationally efficient.

On the side of research ethics, we also believe that we have an obligation to accurately define the model we *actually* used in our code. We can't simply describe the model one way in the text, and describe it in a different way in the code.

Similar computational tricks are seen all over industry. Take for example, the computational problem of calculating the log of the sum of exponentials. If computer scientists were to write code the way such a statement is typically described verbally: $\log(\text{sum}(\exp(\text{Alpha})))$, where Alpha is a vector, software that uses such an implementation is prone to catastrophic failure due to numerical issues with floating point arithmetic.

Social scientists might be inconvenienced by having to learn a bit of new notation, but we believe that that it is better than having numerical bugs in our models. For social scientists, numerical bugs might be a bit of an inconvenience, but similar mistakes in other fields are treated much more seriously, because numerical issues in computers running medical equipment, airplane controls, or similar systems are no small problem. Engineers are therefore much more likely to emphasize writing and describing computational models in ways that are numerically stable⁸, we argue that social scientists should hold ourselves to the same standards.

For these reasons, we keep our description of Eq. (4) in our manuscript as it is currently written. As an aside, it is not **awkward**, it is an **incredibly elegant, clever** formula, one that resulted from cutting edge research by Lewandowski et al. (2009) and others, and a careful reading of the statistical literature by those on the Stan development team, to develop and promote better ways of formulating priors for multi-level models.

Also, the * symbol is undefined.

The * symbol was just there to show multiplication. We now define the * symbol as indicating the standard matrix product.

⁶ Gelman, Andrew. "Prior distributions for variance parameters in hierarchical models (comment on article by Browne and Draper)." (2006): 515-534.

⁷ Lewandowski, D., Kurowicka, D., & Joe, H. (2009). Generating random correlation matrices based on vines and extended onion method. *Journal of multivariate analysis*, 100(9), 1989-2001; Stan User Guide 2023. <https://mc-stan.org/docs/stan-users-guide/reparameterization.html>

⁸ <https://gregorygundersen.com/blog/2020/02/09/log-sum-exp/>

- With regard to P2, I had trouble understanding how gameplay could be anonymous and yet affect dynamic ratings. Doesn't one need a stochastic process model here to understand updating?

Game play **is** anonymous, but it **does not** affect dyadic ratings. We explicitly (e.g., lines 170-182 and 310) state that game play is anonymous, and that people **cannot** use observations of other people's gameplay to base their own economic decisions. This is not an oversight, this is by design. Our predictions here are designed to test the external validity of the RICH games.

We state on lines 175-182:

“...our measures of social standing/reputation are not based on game-play, but rather reflect individuals' pre-existing perceptions of other group members.”

It is hard to see how this statement could be mistaken for suggesting that game-play affects ratings.

We would expect prediction P2 to be satisfied if and only if, individual j's ongoing—that is real-world—perception of individual k as selfish or generous actually tracks how individual k treats individual j on an ongoing basis *in the real world*, AND individual k's behavior towards individual j in the game *accurately proxies for k's behavior towards individual j in the real world*. The fact that we don't reveal private gameplay decisions means that we could only recover statistically reliable results if people's gameplay behavior closely proxies their ongoing behavior in the real world that would lead to them being perceived of as either selfish or generous.

We were already careful to make this clear on lines 210-225 of the Predictions section, stating:

“...perceived standing will be influenced by cooperative behavior, with individual j's perception of individual k tracking how k treats j **on an ongoing basis**. Assuming that **behavior in the economic games parallels or proxies for behavior in real-world contexts** (i.e., **that individual j is more likely to give to individual k in the allocation game if j gives to, or shares with, k in real-world contexts**; see Pisor et al. (2020) for evidence from Colombia), this leads to predictions that..”

and line 790 in the Results section, stating:

“...we show that dyadic perceptions of generosity and selfishness (individual j rating k) seemed to track behavior in the RICH games (individual k transferring to j); in other words, dyadic perceptions of generosity and selfishness tended to accurately track dyadic resource flows in the game, even though economic behavior in the game was private (**speaking to the ecological validity of the games**).”

We make the problem harder for ourselves, and we still find evidence that reputations track behavior. We don't see any need to respond to this point. We already address the reviewer's concerns fully in the manuscript.

- I am so confused about ρ . I thought it was a matrix with dimensionality of J^2 -by- J^2 for each m with a particular structure in (14). Now, starting at line 575, it seems to be a scalar with a posterior mean and CI. I am so lost that I stopped here.

We are confused as to how the reviewer could be confused. Right on line 560 (just a few centimeters away from where the reviewer got lost), we introduce the notation that we will use in the results section, stating:

“We present estimates—posterior means and 89% credible intervals—from the coastal community **using the symbol ρ_c** , the lowland community **using the symbol ρ_l** , the highland community **using the symbol ρ_h** , and from the altiplano community **using the symbol ρ_a** .”

In other words, we tell the reader that we will be using subscripts on ρ to refer to the field-site. And then, on line 567, we direct the reader to Figure 3, stating:

“Fig. 3 plots a matrix of dyadic reciprocity coefficients from the lowland community.”

Figure 3 visualizes the **exact structure** of the matrix ρ_l , which is, exactly as we showed in Eq. 4 and Eq. 14, a $2M \times 2M$ matrix. If the reviewer was confused as to the dimensions of the correlation matrix, he or she could have simply looked at **Figure 3** and counted how long each side of the square was. Because we have $M=5$ network layers (3 games layers and 2 reputation layers), the matrix is 10 by 10.

The reviewer then is confused that we report scalars labeled by ρ throughout the results section. This level of obfuscation seems deliberate. We fail to believe that the reviewer couldn't figure this out from reading the manuscript. It is totally normative for researchers in psychology to refer to a correlation matrix as rho, and then to present its elements in the text using the same symbol rho, as long as they are careful to describe which variables that element of rho corresponds to.

In Figure 3(a), in the second line of the caption, we even teach readers how to interpret our quoted results as elements of the relevant correlation matrices:

“For example, we see that if individual j gave to individual k in the giving game, then individual j is reliably less likely to exploit individual k in the exploitation game ($\rho = -0.62$).”

It's a shame that the reviewer “gave up” at line 575; if they would have turned the page, and read the caption on Figure 3, maybe they could have learned how to interpret the results stemming from Figures 3 and 4. As such, we did not modify the text to deal with this concern of the reviewer, as the reviewer simply refused to read the text that was already written, that would have clarified the issue.

More generally, *a matrix is simply an organized array of scalars*. An $N \times N$ correlation matrix for example, “holds” the pairwise correlations between N different variables. Thus, we can pull out an element from a correlation matrix, and that will be a scalar. Because the correlation matrix is itself an unknown parameter, and the target of our Bayesian estimation procedure, every element in it has a posterior distribution. Those posterior distributions are what we summarize with a posterior mean and CI.

As we said before, we teach readers how to interpret the elements of each ρ matrix, through examples in the figure captions. Obviously, in the main text, we are referring to specific elements in each labeled correlation matrix. For example, if we are talking about the correlation of individual j 's propensity to give to k with individual j 's propensity to exploit k , we present the value of that correlation in a given field-site by pulling the relevant element out of the larger $2M \times 2M$ matrix. Rather than adopting some obtuse and excessively verbose way of labeling our rhos—e.g., $\rho_{[c, j \text{ gives to } k, j \text{ exploits } k]}$ —we keep our

current notational conventions, and note that our prose makes it clear which elements of the matrix we are talking about. We can't imagine many readers being confused by this.

Finally, we noted previously on line 392: "...assume we have collected **M network layers of economic game and peer ratings data...**" and on line 445 (Eq. 3) we show, and describe, concatenating two M-vectors. As such, rho must be a $2M \times 2M$ matrix. There is no ambiguity. Above all, we certainly never said anywhere that the correlation matrix was J^2 -by- J^2 for each m. The text " J^2 " never appears in the paper.

Nevertheless, we do make one small change. We now define Sigma, L, Gamma, and Rho to be of dimension $2M \times 2M$, explicitly, rather than leaving the reader to conclude as much on their own.

Take home: This paper has the potential to succeed as it uses cutting edge methods to address important questions. Nonetheless, I think there are huge gaps. Perhaps I am too far a field to understand this paper. My sense, however, is that many others will struggle. The logic and articulation between question and method is not transparent.

We are certainly open to constructive criticism, and we have modified the paper considerably upon the critical, but good-spirited feedback of the other reviewers. With respect to reviewer 4, however, there is very little we can do, other than rebut their arguments. The models we developed for this paper are substantially more complicated than what is commonly seen in the social science literature, and so some readers may indeed struggle. However, we developed this model specifically to address known issues in analyzing multi-layer networks. Some of these issues, like potential collinearity concerns, are addressed by our more complex model, but would affect lazier approaches (e.g., just including sentiments as predictors of allocation decisions using the basic SRM). In struggling through our approach, we do hope readers will learn something new, both in terms of the content of the study, but perhaps just as importantly, we hope readers learn how to use the multiplex social relations model to analyze multi-layer social network data using cutting edge tools that robustly account for and estimate correlations in random effects both within and between layers.

16th Feb 24

Dear Dr Redhead,

Your manuscript titled "Reputation, exploitation, and the maintenance of cooperation: Evidence of direct and indirect reciprocity in network-structured economic games" has now undergone a final round of peer-review. The comments appear below. In light of the peer reviewer's advice I am delighted to say that we are happy, in principle, to publish a suitably revised version in Communications Psychology under the open access CC BY license (Creative Commons Attribution v4.0 International License).

We also apologize for the significant delay in reaching the decision. Your final revisions dealt with the issues raised by our statistics expert. Unfortunately, they were ultimately unable to provide us with a peer-review report and we needed to recruit a new, fifth reviewer. As you will see, Patricia Lockwood, one of Communications Psychology's editorial board members ultimately kindly offered to review your revision. Please note that Dr Lockwood acted as a referee in this instance, not in her role as an editor. We take the unusual course of action to request a review from an editorial board member when we've exhausted all other routes to solicit expert advice from an independent referee.

We invite you to revise your paper one last time to address the remaining concerns of our reviewers and a list of editorial requests. At the same time we ask that you edit your manuscript to comply with our format requirements and to maximise the accessibility and therefore the impact of your work.

EDITORIAL REQUESTS:

SUBMISSION INFORMATION:

OPEN ACCESS:

Communications Psychology is a fully open access journal. Articles are made freely accessible on publication under a CC BY license (Creative Commons Attribution 4.0 International License). This license allows maximum dissemination and re-use of open access materials and is preferred by many research funding bodies.

For further information about article processing charges, open access funding, and advice and support from Nature Research, please visit <https://www.nature.com/commspsychol/article-processing-charges>

At acceptance, you will be provided with instructions for completing this CC BY license on behalf of all authors. This grants us the necessary permissions to publish your paper. Additionally, you will be asked to declare that all required third party permissions have been obtained, and to provide billing information in order to pay the article-processing charge (APC).

* TRANSPARENT PEER REVIEW: Communications Psychology uses a transparent peer review system. On author request, confidential information and data can be removed from the published reviewer reports and rebuttal letters prior to publication. If you are concerned about the release of confidential data, please let us know specifically what information you would like to have removed. Please note that we cannot incorporate redactions for any other reasons.

* CODE AVAILABILITY: All Communications Psychology manuscripts must include a section titled "Code Availability" at the end of the methods section. We require that the custom analysis code supporting your conclusions is made available in a publicly accessible repository at this stage; please choose a repository that generates a digital object identifier (DOI) for the code; the link to the repository and the DOI must be included in the Code Availability statement. Publication as Supplementary Information will not suffice.

* DATA AVAILABILITY:

[Link redacted]

Best regards,

Antonia Eisenkoeck

Antonia Eisenkoeck
Senior Editor
Communications Psychology

REVIEWERS' COMMENTS:

Reviewer #5 (Remarks to the Author):

The authors have been comprehensive and informative in their reply to the remaining reviewer comments. They have provided further evidence that their model is indeed sound. I believe the manuscript is ready for publication. Many congratulations to the authors!